# TopER: Topological Embeddings in Graph Representation Learning

**Astrit Tola**
Department of Mathematics
Florida State University
Tallahassee, FL 32306
atola@fsu.edu

**Funmilola Mary Taiwo**
Department of Statistics
University of Manitoba
Winnipeg, Manitoba, Canada
taiwom1@myumanitoba.ca

**Cuneyt Gurcan Akcora**
AI Institute
University of Central Florida
Orlando, FL, 32816
cuneyt.akcora@ucf.edu

**Baris Coskunuzer**
Department of Mathematical Sciences
University of Texas at Dallas
Richardson, TX 75080
coskunuz@utdallas.edu

## Abstract

Graph embeddings play a critical role in graph representation learning, allowing machine learning models to explore and interpret graph-structured data. However, existing methods often rely on opaque, high-dimensional embeddings, limiting interpretability and practical visualization.

In this work, we introduce Topological Evolution Rate (TopER), a novel, low-dimensional embedding approach grounded in topological data analysis. TopER simplifies a key topological approach, Persistent Homology, by calculating the evolution rate of graph substructures, resulting in intuitive and interpretable visualizations of graph data. This approach not only enhances the exploration of graph datasets but also delivers competitive performance in graph clustering and classification tasks. Our TopER-based models achieve or surpass state-of-the-art results across molecular, biological, and social network datasets in tasks such as classification, clustering, and visualization.

> 🐍 `pip install toper`

## 1 Introduction

Graphs are a fundamental data structure utilized extensively to model complex interactions within various domains, such as social networks [LBKT08], molecular structures [YLY+18], and transportation systems [DCS+22]. Their inherent flexibility, however, introduces significant challenges when applied to machine learning (ML) tasks, primarily due to their irregular and high-dimensional nature. Graph data lacks inherent ordering and consistent dimensionality, making it challenging for traditional ML methods designed for vector spaces.

Graph Neural Networks (GNNs) have emerged as the state-of-the-art models for tackling graph machine learning tasks due to their ability to learn effectively from graph-structured data [KW17]. In the predominant paradigm of message-passing GNNs, the process begins by generating node embeddings [BHG+21]. These embeddings can then be used in tasks such as node classification or link prediction. However, for graph-related tasks, such as molecular property prediction, the embeddings

39th Conference on Neural Information Processing Systems (NeurIPS 2025).

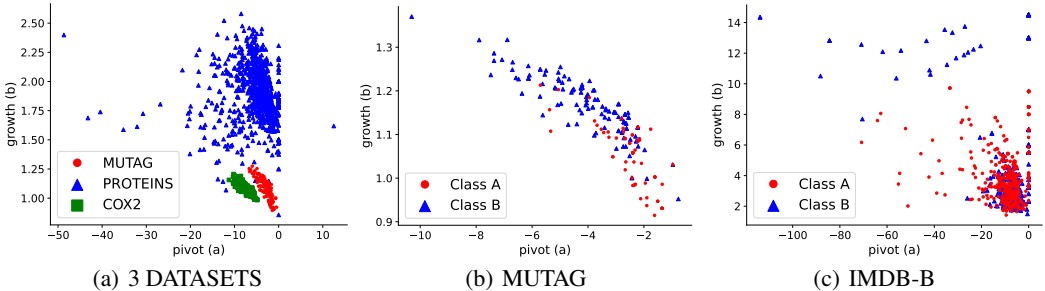

Figure 1: **TopER Visualizations.** Each data point represents an individual graph. On the left, TopER is applied to three benchmark compound datasets using closeness sublevel filtration. The middle panel zooms in on the red point cloud from the left, demonstrating TopER's effectiveness in distinguishing between classes within the MUTAG dataset. On the right, a TopER visualization for the IMDB-B dataset is displayed.

must be aggregated through a pooling layer to form graph-level representations [WKK+20]. This method is computationally intensive, mainly because generating and managing node embeddings as intermediate steps substantially increases the overall computational burden. Ideally, an approach would allow for the direct creation of graph embeddings, circumventing the need to generate node-level representations first. Furthermore, these graph embeddings must be both low-dimensional and interpretable to maximize their practical utility and efficiency in various applications.

Topological Data Analysis (TDA) is well-suited for directly constructing graph representations without costly node embeddings. Topology studies the shape of data, and TDA primarily focuses on the qualitative properties of space, such as continuity and connectivity [CA24]. A particularly effective technique in TDA is *Persistent Homology (PH)*, which tracks topological features, like connected components and cycles, across various scales via a process known as filtration. Filtration is adept at revealing both local and global structures within graphs. It proves exceptionally useful for comparing graphs of different sizes that maintain the same inherent structure, which may suggest similar properties in graph datasets. For instance, similar substructures in protein interaction networks across different species may indicate comparable biological functions. By focusing on data shape, PH proves invaluable in graph tasks that benefit from a graph-centric approach, offering insights that might not be as apparent when focusing on individual node analysis. This shift from a node-centric to a graph-centric perspective can dramatically improve the understanding and application of graph data in fields like bioinformatics and network analysis. However, the utility of Persistent Homology is limited by the high computational demands involved in extracting topological features during the filtration process, mainly due to its cubic time complexity [OPT+17]. This constraint reduces its practicality for large-scale graphs and has restricted the broader integration of PH in graph representation learning.

With this work, we take a significant step forward in addressing the challenges of topological graph representation learning and introduce *Topological Evolution Rate (TopER)*. This novel approach refines the Persistent Homology process to efficiently capture graph substructures, thereby mitigating the significant computational demands of calculating complex topological features. As graph representation learning aligns naturally with Topological Data Analysis, *TopER* excels in graph clustering and classification tasks, where it achieves the best rank in experiments. Furthermore, simplifying graph data into a low-dimensional space, *TopER* creates intuitive visualizations that reveal clusters, outliers, and other essential topological features, as demonstrated in Figure 1. As a result, *TopER* merges interpretability with efficiency in graph representation learning, providing an ideal balance that can scale to large graphs. To our knowledge, *TopER* is the first topology-based graph representation method that can create low-dimensional, efficient, and scalable graph representations.

Our contributions can be summarized as follows:

- **New Graph Representation:** We introduce *TopER*, a principled and computationally feasible graph representation designed to capture the structural evolution of a graph.

- **Topology-Inspired Design:** Rather than computing persistent diagrams or Betti numbers, *TopER* directly leverages the filtration process to provide efficient, low-dimensional summaries rooted in topological intuition.

- **Competitive Performance:** Experiments on benchmark graph classification and clustering tasks demonstrate that *TopER* achieves competitive results compared to more complex, state-of-the-art models, while offering superior interpretability.

- **Interpretable Visualizations:** *TopER* produces low-dimensional outputs that support intuitive visualization of clusters and structural outliers within and across graph datasets, enabling a form of visual model comparison often lacking in traditional embeddings.

- **Robustness Under Perturbations:** We provide a stability analysis showing that *TopER*'s representations are consistent under small changes to the filtration function, reinforcing its suitability for practical and comparative applications.

## 2 Background

### 2.1 Related Work

**Graph-level Embedding Methods.** Graph representation learning, including GNNs and Graph Pooling techniques, is a dynamic subfield of machine learning, focusing on transforming graph data into efficient, low-dimensional vector representations that encapsulate essential features of the data [Ham20, GHT+19]. These representations facilitate a deeper analytical understanding of graphs, which is critical for various applications such as molecular graph property prediction [DTRF19]. GNNs have revolutionized the analysis of graph data, drawing parallels with the success of Convolutional Neural Networks in image processing [EPBM20]. GNNs utilize spectral and spatial approaches to graph convolutions based on the graph Laplacian and direct graph convolutions, respectively [BZSL14, DBV16, KW17]. Despite their success, GNNs often suffer from issues like over-smoothing and a lack of transparency, making them less ideal for applications requiring interpretability [Gün22].

Graph Pooling emerged as a key component in GNN architectures, drawing parallels to the role of pooling in Convolutional Neural Networks [EPBM20]. Pooling aims to deduce into meaningful graph embeddings through node aggregation (mean, max, and add pooling [XHLJ19]) or hierarchical pooling (node selection: Top-k [GJ19] and SAGPool [LLK19]; node clustering: DiffPool [YYM+18] and MinCutPool [BGA19]). Both pooling groups have their challenges. Node aggregation methods may lose the structural information by treating each node equally, while the hierarchical pooling methods can be computationally heavy, and loss of information can occur if important nodes are discarded.

**Topology-Inspired Representations.** TDA provides a robust and computationally efficient framework to address the interpretability and over-smoothing issues present in GNNs [AAF19]. Persistent Homology, a key technique in TDA, has been applied successfully to graph data, demonstrating potential to match or even exceed the performance of traditional methods in classification and clustering tasks [HMR21, DCG+22, HIL+24, ISG23, CSA+23, LSC+23]. However, the computational intensity of PH limits its scalability [HKN19, ZYCW20, AKGC22].

**Visualization Techniques.** Graph embedding techniques, including spectral methods, random walk-based approaches, and deep learning-based models, transform graph data into vector representations to support tasks like visualization and machine learning [CZC18, GF18, Xu21]. Approaches such as Laplacian Eigenmaps and DeepWalk have been particularly effective in revealing clusters within graphs [BN01, PAS14]. However, these methods are predominantly applied to visualize a *single graph* in node classification tasks, focusing on cluster identification [WCZ16, MK20, TPPM23]. Furthermore, they often overlook domain-specific information, which can limit their effectiveness in more specialized applications [JZ20].

*TopER* addresses these challenges by combining the interpretative benefits of TDA with the analytical strength of modern graph machine learning. Distinct from current approaches, TopER employs a simplified filtration process to create embeddings that are both interpretable and computationally efficient. By extending the filtration to multiple functions, TopER stands out as one of the first methods to offer effective and interpretable visualizations of graph datasets, while also achieving superior performance in clustering and classification tasks.

## 2.2 Persistent Homology for Graphs

Topological Data Analysis (TDA) offers a powerful framework for graph representation learning [AAF19], with persistent homology (PH) being especially effective at capturing multi-scale topological features [CA24]. While PH typically involves filtrations, persistence diagrams, and vectorization, our model focuses on the filtration step, reformulating the evolution of topological features in a novel and efficient way.

In the crucial filtration step, PH decomposes a graph $\mathcal{G}$ into a nested sequence of subgraphs $\mathcal{G}_1 \subseteq \mathcal{G}_2 \subseteq \ldots \subseteq \mathcal{G}_n = \mathcal{G}$. For each $\mathcal{G}_i$, an abstract simplicial complex $\widehat{\mathcal{G}}_i$ is defined, forming a filtration of simplicial complexes. Clique complexes are typical choices, where each $(k+1)$-complete subgraph in $\mathcal{G}$ corresponds to a $k$-simplex [AAF19].

**Filtration.** Utilizing relevant filtration functions is essential to obtain effective filtrations. For a given graph $\mathcal{G} = (\mathcal{V}, \mathcal{E})$, a common approach is to define a node filtration function $f : \mathcal{V} \to \mathbb{R}$, which establishes a hierarchy among the nodes. By selecting a monotone increasing set of thresholds $\mathcal{I} = \{\epsilon_i\}_{i=1}^n$, this method generates subgraphs $\mathcal{G}_i = (\mathcal{V}_i, \mathcal{E}_i)$ where $\mathcal{V}_i = \{v \in \mathcal{V} \mid f(v) \le \epsilon_i\}$ and $\mathcal{E}_i$ is the set of edges in $\mathcal{E}$ with endpoints in $\mathcal{V}_i$. This is called a sublevel filtration induced by $f$ (See Figure 2). Also, superlevel filtrations can be constructed by defining $\mathcal{V}_i = \{v \in \mathcal{V} \mid f(v) \ge \epsilon_i\}$ for decreasing thresholds [AAF19].

Similarly, one can use edge filtration functions $g : \mathcal{E} \to \mathbb{R}$ to define such a filtration. We define $\mathcal{E}_i = \{e_{jk} \in \mathcal{E} \mid g(e_{jk}) \le \epsilon_i\}$, and $\mathcal{V}_i$ as all the endpoints of $\mathcal{E}_i$ to create a nested sequence $\{\mathcal{G}_i\}_{i=1}^n$. Especially for weighted graphs, this method is highly preferable as weights naturally define an edge filtration function. The common node filtration functions are degree, betweenness, centrality, heat kernel signatures [BK10], and node functions coming from the domain of the datasets (e.g., atomic number for molecular graphs). Common edge filtration functions are Ollivier and Forman Ricci curvatures [LLY11] and edge weights (e.g., transaction amounts for financial networks).

In addition to existing approaches, we introduce a new filtration function, *Popularity*, which extends the idea of degree-based ranking [New03]. While the degree function measures the number of direct neighbors a node has, *Popularity* captures the average degree of those neighbors. The underlying intuition is that, whereas degree reflects how many connections a node has, *Popularity* emphasizes node influence through association with high-degree nodes.

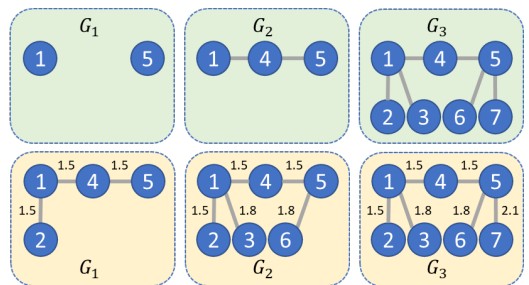

Figure 2: **Filtration.** For $\mathcal{G} = \mathcal{G}_3$ in both examples, the top figure illustrates superlevel filtration with node degree function for thresholds $\{1, 2, 3\}$. Similarly, the bottom figure illustrates sublevel filtration for edge weights with thresholds $\{1.5, 1.8, 2.1\}$.

Formally, for each node $v$, we define popularity as: $\mathcal{P}(v) = \deg(v) + \frac{\sum_{u \in \mathcal{N}(v)} \deg(u)}{|\mathcal{N}(v)|}$ where $\deg(v)$ is the node's degree and $\mathcal{N}(v)$ is its set of neighbors. This function incorporates 2-neighborhood information by weighting high-degree neighbors more heavily, making it a refined version of the degree function.

## 3 TopER: Topological Evolution Rate

TopER is inspired by the foundational idea that topology can capture the shape of graph data, and that this shape can be observed through the evolution of a graph during the filtration process. In this sense, TopER is topology-inspired. However, TopER diverges from traditional PH in how it tracks the shape. We first achieve a computationally efficient alternative to persistent homology by simplifying its filtration-based perspective, and second, we develop a low-dimensional, interpretable representation of graphs that enables both effective classification and intuitive visualization across graph datasets.

Our reformulation reduces the computational overhead typically required for topological feature extraction. Unlike Persistent Homology, which extracts costly topological features, TopER summarizes the filtration process through two key parameters derived via regression: *filtration sequences* and *evolution*.

**Filtration Sequences.** We first decompose a graph $\mathcal{G}$ into a nested sequence of subgraphs (filtration graphs) $\mathcal{G}_1 \subseteq \mathcal{G}_2 \ldots \subseteq \mathcal{G}_n = \mathcal{G}$ by using a filtration function, such as node degree or closeness. Let $\mathcal{G}_i \subset \mathcal{G}$, $\mathcal{V}_i$ represent nodes in $\mathcal{G}_i$ and $\mathcal{E}_i$ represent the edges. Next, we compute $x_i = |\mathcal{V}_i|$ as the count of nodes, and $y_i = |\mathcal{E}_i|$ as the count of edges. Then, for each filtration graph $\mathcal{G}_i$, we obtain the pair $(x_i, y_i) \in \mathbb{R}^2$, which creates two monotone sequences, $x_1 \leq x_2 \leq \cdots \leq x_n$ and $y_1 \leq y_2 \leq \cdots \leq y_n$. Hence, TopER yields two ordered sets $\mathcal{X}, \mathcal{Y}$ describing the evolution of the filtration graphs $\mathcal{G}_1 \subseteq \ldots \subseteq \mathcal{G}_n = \mathcal{G}$, $\mathcal{X} = (x_1, x_2, \ldots, x_n)$ and $\mathcal{Y} = (y_1, y_2, \ldots, y_n)$. Here, $n$ corresponds to number of thresholds $\{\epsilon_i\}_{i=1}^n$ used in the filtration step. Consider the top row of Figure 2 where we have three filtration graphs (i.e., $n$=3); we have $\mathcal{X} = (2, 3, 7)$ for node counts and $\mathcal{Y} = (0, 2, 6)$ for edge counts.

**Evolution.** In the next step, PH would typically compute topological features on each filtration and create a persistence diagram to summarize the features. Not only is it costly, but the approach would require efforts to vectorize the persistence diagrams. We circumvent this computationally costly step and analyze how the number of edges $\{y_i\}$ relates to the number of nodes $\{x_i\}$ throughout the filtration sequence. We use line fitting to characterize this relationship as follows. Simple linear regression, often applied through the least

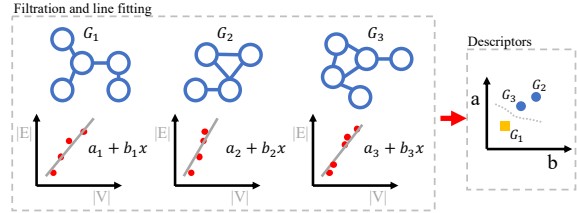

Figure 3: **TopER steps.** The filtration process on three different graphs using node or edge filtration. The graphs undergo filtration, and for each graph, a best-fit line is determined through the filtration data. The coefficients of these best-fit lines are then used as descriptors for the graphs.

squares method [JWH$^+$23], is a standard approach in regression analysis for fitting a linear equation to a set of data points $\{(x_i, y_i)\} \subset \mathbb{R}^2$. This method calculates the line $L(x) = a + bx$ that best fits the data by minimizing the loss function $\mathrm{E} = \sum_{i=1}^N [L(x_i) - y_i]^2$. The regression coefficients $(a, b)$ capture the graph's structural evolution through filtration (see descriptor step in Figure 3). The full TopER method is outlined in Algorithm 1 in the Appendix.

With evolution on filtration sequences, we define the topological evolution rate of a graph as follows:

**Definition 3.1** (Topological Evolution Rate (TopER)). Let $f : \mathcal{V} \to \mathbb{R}$ be a filtration function on graph $\mathcal{G}$ and $\mathcal{I} = \{\epsilon_i\}_{i=1}^n$ be the threshold set. Let $\mathcal{G}_i = (\mathcal{V}_i, \mathcal{E}_i)$ be the induced filtration. Let $x_i = |\mathcal{V}_i|$ and $y_i = |\mathcal{E}_i|$. Let $L(x) = a + bx$ be the best fitting line to $\{(x_i, y_i)\}_{i=1}^n$. Then, we define the *TopER vector* of $\mathcal{G}$ wrt. $f$ as $\mathrm{TE}_f(\mathcal{G}, \mathcal{I}) = (\mathrm{a}, \mathrm{b})$. We call $a$ the *pivot* and $b$ the *growth* of $\mathcal{G}$.

*Remark* 3.2 (Why a Linear Fit?). Although one could consider higher-order polynomials, our experiments show that coefficients beyond the first degree are negligible (Table 15), revealing an essentially linear relationship. Due to space constraints, we defer a more detailed comparison of linear, polynomial, and higher-order fits to Appendix B, where we also provide visual examples of evolution rates (Figure 6) and fundamental pattern types (Figure 7).

*Remark* 3.3 (On the Name *TopER*). Although *TopER* does not compute classical topological invariants such as persistence diagrams or Betti numbers, its name is grounded in two topology-inspired principles. First, it mirrors the TDA notion of a *filtration*, a nested sequence of subgraphs $\{\mathcal{G}_i\}$ that progressively unveils structural features across scales. Second, it is directly tied to the evolution of the Euler characteristic $\chi(\mathcal{G}_i) = |\mathcal{V}_i| - |\mathcal{E}_i| = \beta_0(\mathcal{G}_i) - \beta_1(\mathcal{G}_i)$, where $\beta_i$ denotes the $i$th Betti number. Equivalently, one could track $\{(x_i, y_i - x_i)\} = \{(|\mathcal{V}_i|, \chi(\mathcal{G}_i))\}$, but we use $\{(x_i, y_i)\}$ for notational simplicity. The resulting slope thus quantifies how topological complexity changes over the filtration, justifying the term *Topological Evolution Rate*.

We emphasize that the term "topological" reflects the conceptual roots of TopER in topological data analysis: the method uses a filtration to reveal structural patterns in the graph, analogous to the filtration process in persistent homology. While we do not compute full persistence diagrams or Betti numbers, the slope of the Euler characteristic across the filtration captures essential topological information, justifying the terminology.

### 3.1 Computational Complexity

The primary computational steps in TopER include constructing filtration graphs and performing regression on node and edge counts, which incur the following costs.

Analyzing each node and edge across $n$ filtration thresholds typically requires $O(n \times (|\mathcal{V}| + |\mathcal{E}|))$ operations, where $\mathcal{V}$ and $\mathcal{E}$ denote the numbers of vertices and edges, respectively. The regression step involves fitting a line to the pairs $(x_i, y_i)$ using the least squares method. The complexity of calculating the necessary sums for this regression is $O(n)$, and solving for the regression coefficients (slope and intercept) from these sums involves a constant amount of additional computation.

Thus, the overall complexity of TopER predominantly hinges on the graph filtration process, summing up to $O(n \times (|\mathcal{V}| + |\mathcal{E}|))$ where $|\mathcal{V}| \gg n$. As we will show in the next section, the runtime costs of TopER are notably low, making it practical and efficient for large-scale applications.

## 3.2 Stability Results

This section states our theorems on the stability of TopER. In the following, $\mathcal{W}_p(.,.)$ represents $p$-Wasserstein distance, and $\mathrm{PD}_k(\mathcal{X}, f)$ represents $k^{th}$ persistence diagram of $\mathcal{X}$ with sublevel filtration with respect to $f$. Similarly, $\|.\|_p$ represents $L^p$-norm and $d_p(.,.)$ represents $l_p$-distance in $\mathbb{R}^m$. We fix a threshold set $\mathcal{I} = \{\epsilon_i\}_{i=1}^n$ for both functions to keep the exposition simple. Further, to keep the setting general, we use the pairs $\{(\beta_0(\epsilon_i), \beta_1(\epsilon_i))\}_{i=1}^n$ in $\mathbb{R}^2$ to fit the least squares line $y = a + bx$ defining $\mathrm{TE}(\mathcal{X}) = (a, b)$.

**Theorem 3.4.** *Let $\mathcal{X}$ be a compact metric space, and $f, g : \mathcal{X} \to \mathbb{R}$ be two filtration functions. Then, for some $C > 0$,*

$$\|\mathrm{TE}_f(\mathcal{X}) - \mathrm{TE}_g(\mathcal{X})\|_1 \le C \cdot \mathcal{W}_1(\mathrm{PD}_k(\mathcal{X}, f), \mathrm{PD}_k(\mathcal{X}, g)).$$

By combining the above result with the stability result for sublevel filtrations, we obtain the stability with respect to filtration functions as follows.

**Corollary 3.5.** *Let $\mathcal{X}$ be a compact metric space, and $f, g : \mathcal{X} \to \mathbb{R}$ be two filtration functions. Then, for some $C > 0$,*

$$\|\mathrm{TE}_f(\mathcal{X}) - \mathrm{TE}_g(\mathcal{X})\|_1 \le C \cdot \|f - g\|_1$$

The following lemmas are essential for proving the theorem and corollary above.

**Lemma 3.6.** *[ST20] Let $\mathcal{X}$ be a compact metric space, and $f, g : \mathcal{X} \to \mathbb{R}$ be two filtration functions. Then, for any $p \ge 1$, we have $\mathcal{W}_p(\mathrm{PD}_k(\mathcal{X}, f), \mathrm{PD}_k(\mathcal{X}, g)) \le \|f - g\|_p$*

The next lemma is on the stability of Betti curves by [DG23] [Proposition 1].

**Lemma 3.7.** *[DG23] Let $\beta_k(\mathcal{X})$ is the $k^{th}$ Betti function obtained from the persistence module $\mathrm{PM}_k(\mathcal{X})$.*

$$\|\beta_k(\mathcal{X}) - \beta_k(\mathcal{Y})\|_1 \le 2\mathcal{W}_1(\mathrm{PD}_k(\mathcal{X}), \mathrm{PD}_k(\mathcal{Y}))$$

By adapting the above results to the graph setting, when two metric graphs $\mathcal{G}_1, \mathcal{G}_2$ are close in the Gromov-Hausdorff sense, one can obtain a similar stability result for the filtrations of $\mathcal{G}_i$ induced by the same filtration function. Due to space limitations, details and the proofs are given in Appendix C.

## 4 Experiments

We evaluate the performance of TopER in classification, clustering and visualization. Our Python implementation is available at `https://github.com/AstritTola/TopER`.

### 4.1 Experimental Setup

**Datasets.** We conduct experiments on nine benchmark datasets for graph classification. These are (i)

Table 1: Characteristics of the benchmark graph classification datasets.

| Datasets | #Graphs | $|\mathcal{V}|$ | $|\mathcal{E}|$ | Classes |
|---|---|---|---|---|
| BZR | 405 | 35.75 | 38.36 | 2 |
| COX2 | 467 | 41.22 | 43.45 | 2 |
| MUTAG | 188 | 17.93 | 19.79 | 2 |
| PROTEINS | 1113 | 39.06 | 72.82 | 2 |
| IMDB-B | 1000 | 19.77 | 96.53 | 2 |
| IMDB-M | 1500 | 13.00 | 65.94 | 3 |
| REDDIT-B | 2000 | 429.63 | 497.75 | 2 |
| REDDIT-5K | 4999 | 508.52 | 594.87 | 5 |
| OGBG-MOLHIV | 41127 | 243.4 | 2266.1 | 2 |

the molecule graphs of BZR, and COX2 [MV09]; (ii) the biological graphs of MUTAG and PRO-TEINS [KM12]; and (iii) the social graphs of IMDB-Binary (IMDB-B), IMDB-Multi (IMDB-M), REDDIT-Binary (REDDIT-B), and REDDIT-Multi-5K (REDDIT-5K) [YV15]. Finally, the OGBG-MOLHIV is a large molecular property prediction dataset, part of the open graph benchmark (OGB) datasets [HFZ$^+$20]. Data statistics are given in Table 1.

Table 2: **Graph Classification.** Accuracy results on eight benchmark datasets. Best results are in **bold blue**, second-best are underlined. The final column shows each model's average deviation from the best per dataset.

| Model | BZR | COX2 | MUTAG | PROTEINS | IMDB-B | IMDB-M | REDDIT-B | REDDIT-5K | Avg.↓ |
|---|---|---|---|---|---|---|---|---|---|
| DiffPool [YYM+18] | $83.93_{\pm4.41}$ | $79.66_{\pm2.64}$ | $79.22_{\pm1.02}$ | $73.63_{\pm3.60}$ | $68.60_{\pm3.10}$ | $45.70_{\pm3.40}$ | $79.00_{\pm1.10}$ | – | 8.06 |
| P-WL-C [RBB19] | – | – | $90.51_{\pm1.34}$ | $75.27_{\pm0.38}$ | – | – | – | – | 2.08 |
| SAGPool [LLK19] | $82.95_{\pm4.95}$ | $79.45_{\pm2.98}$ | $76.78_{\pm2.12}$ | $71.86_{\pm0.97}$ | $74.87_{\pm4.09}$ | $49.33_{\pm4.90}$ | $84.70_{\pm4.40}$ | – | 6.61 |
| Top-k [GJ19] | $79.40_{\pm1.20}$ | $80.30_{\pm4.21}$ | $67.61_{\pm3.36}$ | $69.60_{\pm3.50}$ | $73.17_{\pm4.84}$ | $48.80_{\pm3.19}$ | $79.40_{\pm7.40}$ | – | 9.70 |
| 1-GIN (GFL) [HGR+20] | – | – | – | $74.10_{\pm3.40}$ | $74.50_{\pm4.60}$ | $49.70_{\pm2.90}$ | $90.20_{\pm2.8}$ | $55.70_{\pm2.90}$ | 2.09 |
| 6 GNNs [EPBM20] | – | – | $80.42_{\pm2.07}$ | $75.80_{\pm3.70}$ | $71.20_{\pm3.90}$ | $49.10_{\pm3.50}$ | $89.90_{\pm1.90}$ | $56.10_{\pm1.60}$ | 4.13 |
| MinCutPool [BGA19] | $82.64_{\pm5.05}$ | $80.07_{\pm3.85}$ | $79.17_{\pm1.64}$ | $\underline{76.62}_{\pm2.58}$ | $70.77_{\pm4.89}$ | $49.00_{\pm2.83}$ | $87.20_{\pm5.00}$ | – | 5.82 |
| DMP [BCL21] | – | – | $84.00_{\pm8.60}$ | $75.30_{\pm3.30}$ | $73.80_{\pm4.50}$ | $50.90_{\pm2.50}$ | $86.20_{\pm6.80}$ | $51.90_{\pm2.10}$ | 4.20 |
| FC-V [ORB21] | $85.61_{\pm0.59}$ | $81.01_{\pm0.88}$ | $87.31_{\pm0.66}$ | $74.54_{\pm0.48}$ | $73.84_{\pm0.36}$ | $46.80_{\pm0.37}$ | $89.41_{\pm0.24}$ | $52.36_{\pm0.37}$ | 4.01 |
| SubMix [YSK22] | $86.34_{\pm2.00}$ | $\underline{84.68}_{\pm3.70}$ | $80.99_{\pm0.60}$ | $67.80_{\pm2.00}$ | $70.30_{\pm1.40}$ | $46.47_{\pm2.50}$ | – | – | 6.15 |
| G-Mix [HJLH22] | $84.15_{\pm2.30}$ | $83.83_{\pm2.10}$ | $81.96_{\pm0.60}$ | $66.28_{\pm1.10}$ | $69.40_{\pm1.10}$ | $46.40_{\pm2.70}$ | – | – | 6.91 |
| RGCL [LWZ+22] | $84.54_{\pm1.67}$ | $79.31_{\pm0.68}$ | $87.66_{\pm1.01}$ | $75.03_{\pm0.43}$ | $71.85_{\pm0.84}$ | $49.31_{\pm0.42}$ | $90.34_{\pm0.58}$ | $56.38_{\pm0.40}$ | 3.56 |
| AutoGCL [YWH+22] | $86.27_{\pm0.71}$ | $79.31_{\pm0.70}$ | $88.64_{\pm1.08}$ | $75.80_{\pm0.36}$ | $72.32_{\pm0.93}$ | $50.60_{\pm0.80}$ | $88.58_{\pm1.49}$ | $\mathbf{56.75}_{\pm0.18}$ | 3.08 |
| FF-GCN [PAMF23] | $\underline{89.00}_{\pm5.00}$ | $78.00_{\pm8.00}$ | $71.00_{\pm4.00}$ | $62.00_{\pm1.00}$ | $63.00_{\pm8.00}$ | – | – | – | 11.53 |
| WWLS [FHSK23] | $88.02_{\pm0.61}$ | $81.58_{\pm0.91}$ | $88.30_{\pm1.23}$ | $75.35_{\pm0.74}$ | $\mathbf{75.08}_{\pm0.31}$ | $\underline{51.61}_{\pm0.62}$ | – | – | 2.26 |
| EPIC [HLAK24] | $88.78_{\pm2.30}$ | $\mathbf{85.53}_{\pm1.60}$ | $82.44_{\pm0.70}$ | $69.06_{\pm1.00}$ | $71.70_{\pm1.00}$ | $47.93_{\pm1.30}$ | – | – | 4.67 |
| EMP [CSA+23] | – | – | $88.79_{\pm0.63}$ | $72.78_{\pm0.54}$ | $74.44_{\pm0.45}$ | $48.01_{\pm0.42}$ | $\underline{91.03}_{\pm0.22}$ | $54.41_{\pm0.32}$ | 2.97 |
| MP-HSM [LSC+23] | – | $77.10_{\pm3.00}$ | $85.60_{\pm5.30}$ | $74.60_{\pm2.10}$ | $\underline{74.80}_{\pm2.50}$ | $47.90_{\pm3.20}$ | – | – | 4.67 |
| TopoGCL [CFG24] | $87.17_{\pm0.83}$ | $81.45_{\pm0.55}$ | $90.09_{\pm0.93}$ | $\mathbf{77.30}_{\pm0.89}$ | $74.67_{\pm0.32}$ | $\mathbf{52.81}_{\pm0.31}$ | $90.40_{\pm0.53}$ | – | $\underline{1.76}$ |
| PGOT [QTLL24] | $87.32_{\pm3.90}$ | $82.98_{\pm5.21}$ | $\mathbf{92.63}_{\pm2.58}$ | $73.21_{\pm2.59}$ | $62.90_{\pm3.05}$ | $51.33_{\pm1.76}$ | – | – | 3.85 |
| RePHINE [ISG23] | – | – | – | $71.25_{\pm1.60}$ | $69.40_{\pm3.78}$ | – | – | – | 5.86 |
| GPSE [CLL+24] | $80.49_{\pm4.18}$ | $78.37_{\pm2.62}$ | $87.19_{\pm8.66}$ | $72.15_{\pm3.66}$ | $69.30_{\pm3.61}$ | $47.40_{\pm5.40}$ | $80.40_{\pm3.40}$ | – | 7.27 |
| **TopER** | $\mathbf{90.13}_{\pm4.14}$ | $82.01_{\pm4.59}$ | $\underline{90.99}_{\pm6.64}$ | $74.58_{\pm3.92}$ | $73.20_{\pm3.43}$ | $50.00_{\pm4.02}$ | $\mathbf{92.70}_{\pm2.38}$ | $56.51_{\pm2.22}$ | $\mathbf{1.60}$ |

**Hardware.** We ran experiments on a single machine with 12th Generation Intel Core i7-1270P vPro Processor (E-cores up to 3.50 GHz, P-cores up to 4.80 GHz), and 32GB of RAM (LPDDR5-6400MHz).

**Runtime.** TopER is highly scalable and can be applied to a 100K node graph in less than 2 minutes (see Figure 5). Our small network experiments took about two days in a shared resource setting, whereas the OGBG-MOLHIV experiments took 7.85 hours. One of the most demanding datasets, REDDIT-5K, requires 2.91 hours to calculate all node and edge functions. The runtime of our methods is dominated by the computation of node functions such as closeness and Riccis blue(see Appendix A.6). Using approximate values for centrality metrics instead could greatly decrease computation time [BP07]. Since this is not our current focus, we leave it as future work.

**Model Setup and Metrics.** We employ a rigorous experimental setup to ensure a fair comparison and the selection of the best graph classification model. We begin by applying BatchNormalization to the input features to maintain consistent scaling. We employ a 90/10 train-test split, adopt the StratifiedkFold strategy, and present the average accuracy from ten-fold cross-validation across all our models. We employ accuracy as the evaluation metric, a widely utilized performance measure within graph classification tasks [EPBM20].

**Filtration Functions.** In TopER, we use both node and edge filtrations (Definition 3.1). Alongside popularity, we apply degree, closeness, and degree centrality [EC22] as node filtration functions and Forman- and Ollivier-Ricci functions [LLY11] as edge filtration functions. We also use atomic weight as a node function for molecular and biological datasets (BZR, COX2, and MUTAG), and node attributes (PROTEINS). We utilized the t-test to assess the statistical significance of each function and applied the Lasso method for regularization. Functions were retained in the model only if they achieved p-values less than 0.05 in the t-test and had non-zero coefficients in the Lasso model [JWH+23]. This approach ensures that the selected filtration functions contribute statistically significant and regularized features to the model. Incorporating additional filtration functions can enhance TopER's ability to analyze graphs from diverse perspectives. However, as we will next illustrate in Table 5, TopER demonstrates strong performance even in its most basic form using the simple and scalable node degree function. This balance of performance and simplicity suits our scalability philosophy; we avoid complex and costly schemes for learning dataset-specific activation functions and homogenize the filtration step in all datasets.

**Classifier.** We utilize a Multilayer Perceptron (MLP) in our graph classification task. The hyperparameters are detailed in Appendix A.7.

## 4.2 Graph Classification Results

**Baselines.** We compare our method with 22 state-of-the-art and recent models in graph classification, including variants of graph neural networks: six GNNs including GCN, DGCNN, Diffpool, ECC, GIN, GraphSAGE which are compared in [EPBM20] (best results of these six GNNs are given in the *6 GNNs row*), FF-GCN [PAMF23]; topological methods: DMP [BCL21], FC-V [ORB21], WWLS [FHSK23], MP-HSM [LSC$^+$23] and EMP [CSA$^+$23]; GNNs enhanced with data augmentation methods: SubMix [YSK22], G-Mix [HJLH22], and EPIC [HLAK24]; GNNs enhanced with contrastive learning methods: RGCL [LWZ$^+$22], AutoGCL [YWH$^+$22], TopoGCL [CFG24]; prototype-based methods: PGOT [QTLL24]; pooling methods: Top-k [GJ19], SAGPool [LLK19], DiffPool [YYM$^+$18], MinCutPool [BGA19] and structural encoder: GPSE [CLL$^+$24].

Table 2 shows the accuracy results for the given models. We use the reported results in the corresponding references for each model. "−" entries in the table mean the reference did not report any result for that dataset. In [EPBM20], the authors compare the six most common GNNs on the graph classification task (see the GNNs row). The last column summarizes each model's overall performance. We report the average of the differences between each model's performance and the best performance in the column across all datasets. If a model's performance is missing for a dataset, it is excluded from the average computation for the model.

Out of eight datasets, TopER achieves the best results in two and ranks second in two other datasets. For the remaining four datasets, TopER's performance is within 4% of the SOTA results. *For overall performance, TopER outperforms all other models with an average deviation of 1.60% from the best performances.* The closest competitor is TopoGCL, which has an average deviation of 1.76%.

**OGBG-MOLHIV results.** To evaluate our model's performance on large datasets, we compare it with recently published models on the OGBG-MOLHIV dataset, as shown in Table 3. The performances of these models are listed in chronological order based on their publication dates, with baseline performances reported from [CPWC23, YCL$^+$21] or the respective model's references. In Appendix A.1, we give further details for TopER performance and the contribution of each function on this dataset. TopER achieves the second-best result on the MOLHIV dataset, while the top-performing model requires learning a significantly larger model with 119.5 million parameters.

**TopER vs. PH.** TopER consistently outperforms Persistent Homology methods in both accuracy and computational efficiency. As shown in Table 4, we compare against the best PH results reported in [Cai21], which evaluates 16 combinations of four filtration functions and four vectorization techniques per dataset. *TopER achieves higher accuracy on all six benchmarks.* In terms of runtime, *TopER is over 10 times faster than PH* on large graphs like Reddit-5K, while maintaining high performance (see Table 8 and Appendix A.2).

Table 3: AUC results for OGBG-MOLHIV dataset.

| Model | AUC |
|---|---|
| GIN-VN [XHLJ19] | $77.80_{\pm1.82}$ |
| HGK-WL [TGL$^+$19] | $79.05_{\pm1.30}$ |
| WWL [BGL$^+$20] | $75.58_{\pm1.40}$ |
| PNA [CCB$^+$20] | $79.05_{\pm1.32}$ |
| DGN [BPL$^+$21] | $79.70_{\pm0.97}$ |
| GraphSNN [WW22] | $79.72_{\pm1.83}$ |
| GCN-GNorm [CLX$^+$21] | $78.83_{\pm1.00}$ |
| Graphormer [YCL$^+$21] | $\mathbf{80.51_{\pm0.53}}$ |
| Cy2C-GCN [CPWC23] | $78.02_{\pm0.60}$ |
| GAWL [NV23] | $78.34_{\pm0.39}$ |
| LLM-GIN [ZZM24] | $79.22_{\pm\text{NA}}$ |
| GMoE-GIN [WJY$^+$23] | $76.90_{\pm0.90}$ |
| TopER | $80.21_{\pm0.15}$ |

Table 4: Accuracy results for TopER vs. Persistent Homology in graph classification tasks.

| | BZR | COX2 | PROTEINS | IMDB-B | IMDB-M | RED-5K |
|---|---|---|---|---|---|---|
| **PH** | $88.4_{\pm0.6}$ | $\mathbf{82.0_{\pm0.6}}$ | $74.0_{\pm0.4}$ | $69.5_{\pm0.5}$ | $46.5_{\pm0.3}$ | $54.1_{\pm0.1}$ |
| **TopER** | $\mathbf{90.1_{\pm4.1}}$ | $\mathbf{82.0_{\pm4.6}}$ | $\mathbf{74.6_{\pm3.9}}$ | $\mathbf{73.2_{\pm3.4}}$ | $\mathbf{50.0_{\pm4.0}}$ | $\mathbf{56.5_{\pm2.2}}$ |

**Ablation Studies.** We have conducted **three ablation studies**. In the first one, we evaluated *the individual performance of each function* as well as their combined effect on classification. As shown in Table 5, the common filtration functions we employ from TDA exhibit strong individual performance. Moreover, when combined, they synergistically enhance overall performance. This is not surprising, as different filtration functions, such as atomic weight or Ricci curvature, generate distinct hierarchies and node-edge distributions, resulting in diverse connectivity patterns throughout the filtration sequence. This diversity is analogous to viewing an object from multiple angles. Hence,

integrating these complementary perspectives improves performance by offering a richer and more varied representation of the graph structure, allowing the model to capture more intricate features. The other two ablation studies are provided in the Appendix A.4, which examines *the effect of the number of thresholds* on TopER's performance, while Appendix A.5 analyzes *the impact of the number of filtration functions* used.

Table 5: **Ablation Study.** Individual and altogether performances of filtration functions with TopER.

| Datasets | Degree-cent. | Popularity | Closeness | Degree | F. Ricci | O. Ricci | Atom weight | TopER |
|---|---|---|---|---|---|---|---|---|
| BZR | $82.22_{\pm 2.13}$ | $82.20_{\pm 3.42}$ | $81.48_{\pm 1.99}$ | $\underline{82.73_{\pm 2.12}}$ | $80.75_{\pm 1.73}$ | $80.99_{\pm 1.48}$ | $82.23_{\pm 2.12}$ | $\mathbf{90.13_{\pm 4.14}}$ |
| COX2 | $\underline{75.38_{\pm 3.96}}$ | $69.21_{\pm 8.19}$ | $67.90_{\pm 7.96}$ | $73.88_{\pm 5.02}$ | $70.46_{\pm 7.28}$ | $73.03_{\pm 4.21}$ | $69.82_{\pm 8.27}$ | $\mathbf{82.01_{\pm 4.59}}$ |
| MUTAG | $76.61_{\pm 7.87}$ | $77.66_{\pm 6.12}$ | $80.88_{\pm 4.79}$ | $74.97_{\pm 6.40}$ | $80.85_{\pm 9.25}$ | $\underline{82.46_{\pm 7.84}}$ | $73.45_{\pm 8.01}$ | $\mathbf{90.99_{\pm 6.64}}$ |
| PROTEINS | $67.66_{\pm 3.16}$ | $70.71_{\pm 4.41}$ | $69.01_{\pm 4.24}$ | $69.01_{\pm 3.48}$ | $72.96_{\pm 3.47}$ | $71.25_{\pm 2.66}$ | $\underline{73.59_{\pm 3.33}}$ | $\mathbf{74.58_{\pm 3.92}}$ |
| IMDB-B | $73.00_{\pm 4.49}$ | $71.90_{\pm 3.48}$ | $72.60_{\pm 4.20}$ | $\underline{73.10_{\pm 4.18}}$ | $69.80_{\pm 2.44}$ | $66.40_{\pm 3.35}$ | - | $\mathbf{73.20_{\pm 3.43}}$ |
| IMDB-M | $\underline{48.47_{\pm 3.90}}$ | $47.87_{\pm 3.07}$ | $48.33_{\pm 3.49}$ | $47.93_{\pm 2.88}$ | $48.13_{\pm 4.11}$ | $43.60_{\pm 3.17}$ | - | $\mathbf{50.00_{\pm 4.02}}$ |
| REDDIT-B | $76.70_{\pm 3.69}$ | $79.35_{\pm 3.46}$ | $78.10_{\pm 3.23}$ | $\underline{79.55_{\pm 2.20}}$ | $72.35_{\pm 2.91}$ | $68.20_{\pm 2.28}$ | - | $\mathbf{92.70_{\pm 2.38}}$ |
| REDDIT-5K | $42.85_{\pm 1.74}$ | $\underline{50.87_{\pm 2.63}}$ | $50.03_{\pm 1.49}$ | $47.01_{\pm 1.89}$ | $50.27_{\pm 1.92}$ | $45.81_{\pm 2.08}$ | - | $\mathbf{56.51_{\pm 2.22}}$ |

### 4.3 Graph Clustering Results

We employ cluster quality metrics to assess the embeddings of graphs sourced from all datasets in Table 2. The embeddings are labeled with their respective dataset memberships, and we assume that good embeddings will have graphs of the same dataset clustered together. We evaluate embeddings based on three widely used clustering metrics: Silhouette (SILH), Calinski-Harabasz (CH), and Davies-Bouldin (DB) [GBC21]. Table 6 compares the clustering performance of TopER and Spectral Zoo [JZ20], which is, to our knowledge, the only model that allows low-dimensional graph embeddings. Detailed results are provided in Appendix A.3. The findings demonstrate that the embeddings generated by TopER outperform those created by Spectral Zoo. This is evident from the superior cluster quality metrics observed for five out of eight datasets in the case of Silhouette and CH, and for all eight datasets in the case of DB.

Table 6: **Clustering Performances.** Comparison of Spectral Zoo vs. TopER. The detailed results are given in Appendix A.3.

| Metric | Method | BZR | COX2 | MUTAG | PROT. | IMDB-B | IMDB-M | REDD-B | REDD-5K |
|---|---|---|---|---|---|---|---|---|---|
| **Silh ↑** | Spec. Zoo | 0.050 | 0.049 | **0.344** | 0.050 | **0.097** | **-0.024** | 0.108 | -0.121 |
| | TopER | **0.249** | **0.414** | 0.258 | **0.086** | 0.064 | -0.032 | **0.196** | **-0.067** |
| **CH ↑** | Spec. Zoo | 3.51 | 6.13 | **120.73** | 38.77 | **85.24** | **30.98** | 269.94 | 119.81 |
| | TopER | **42.58** | **26.00** | 72.52 | **151.64** | 60.52 | 11.77 | **446.12** | **1209.95** |
| **DB ↓** | Spec. Zoo | 7.25 | 6.07 | 0.95 | 4.55 | 2.78 | 10.73 | 2.20 | 25.74 |
| | TopER | **1.93** | **2.29** | **0.88** | **1.54** | **2.19** | **6.87** | **1.32** | **2.78** |

### 4.4 Graph Visualization

In the case of a single filtration function, TopER creates $2D$ graph embeddings $(a, b)$ that can be visualized with ease (see Figure 1). Traditional dimensionality reduction techniques such as PCA can be used to visualize point cloud data, but accurately depicting graph data has historically been a significant challenge [GBGA20]. To our knowledge, until TopER, the only model that allowed graph visualization was the GraphZoo [JZ20].

TopER creates highly interpretable graph visualizations. To recall, the pair $(a, b)$ represents the coefficients of the best-fitting function $L(x) = a + bx$, where $a$ is the pivot (y-intercept) and $b$ is the growth (the slope). Specifically, the pivot $a$ reflects graph connectivity, while $b$ reflects the growth rate of edges/nodes for the filtration function. In particular,

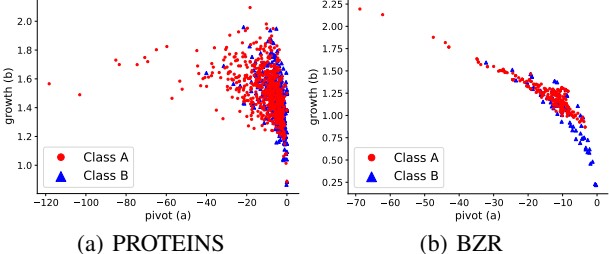

(a) PROTEINS          (b) BZR

Figure 4: TopER visualizations of the PROTEINS dataset with O.Ricci edge filtration, and the BZR dataset with degree centrality node filtration. Each point corresponds to an individual graph.

a higher value of $a$ corresponds to a more interconnected graph. As we demonstrate in Appendix Figure 7, graph connectivity and community structure can be analyzed using three types of pivot behavior. In the following, we illustrate how these quantities can be employed to interpret our two-dimensional representations of the graph datasets.

In Figure 1(b) of the MUTAG dataset, class B has a higher growth rate and smaller pivot than the red class. This shows that the class is growing faster than class A with respect to the closeness function in the MUTAG dataset, i.e., the graph has a low diameter. Similarly, in contrast, in the PROTEINS dataset (Fig. 4(a)), the growth rates are similar for both classes ($\sim 1.5 - 1.7$), but the pivot (initial graph size) is smaller in class A. This implies that class A has fewer edges in relation to the number of nodes. Such patterns, as described in Sec. B.4, can reveal key insights into graph topology. In a similar vein, TopER visualizations can be used for anomaly detection. For example, in Fig. 1(a), an outlier PROTEINS graph alone has a positive pivot and appears as the rightmost data point.

More importantly, **TopER homogenizes graph representations, allowing us to compare graphs across datasets**, which may open new paths in training graph foundation models. To our knowledge, TopER is unique in directly producing interpretable 2D embeddings for cross-dataset visualization without relying on learned high-dimensional encodings or opaque projections, unlike GPSE [CLL+24] and GFSE [CZW+25], which rely on learned high-dimensional embeddings. For example, Figure 1(a) visualizes graphs of three datasets on the same panel, where we see that Mutag and COX2 differ in their pivot only. The similarity is not surprising; MUTAG and COX2 are datasets of molecular graphs where nodes are atoms and edges are chemical bonds. As the molecules in both datasets have similar types of atoms and bond configurations (e.g., ring structures), TopER captures these similarities, leading to similar embeddings.

These examples highlight that, in many practical settings, TopER's interpretability outweighs modest performance differences compared to more expressive or data-driven models. As shown in Figure 4, the pivot–growth representation captures fine-grained structural variations, such as changes in edge density, community organization, and filtration behavior, through simple geometric patterns that can be directly visualized and interpreted. This enables users to pinpoint the topological mechanisms responsible for observed differences between classes or datasets. Beyond classification, such interpretability makes TopER particularly well-suited for applications where structural understanding is critical. Its training-free and computationally efficient design further allows deployment in large-scale or data-scarce environments, as well as integration into hybrid pipelines where TopER embeddings serve as interpretable anchors for downstream learning models. Thus, while deep or spectral approaches may achieve marginally higher benchmark performance in some cases, TopER offers a complementary framework that prioritizes clarity, scalability, and theoretical grounding in topological structure.

**Limitations.** While our approach is designed to be efficient and broadly applicable, its performance can vary depending on the choice of filtration function, which may require domain knowledge in certain applications. In practice, we found the method to be robust across a variety of datasets, and further refinements to filtration strategies could enhance adaptability to new domains.

## 5 Conclusion

We have introduced a novel graph embedding method, *TopER*, leveraging Persistent Homology from Topological Data Analysis. *TopER* demonstrates strong performance in graph classification tasks, rivaling SOTA models. Furthermore, it naturally generates effective 2D visualizations of graph datasets, facilitating the identification of clusters and outliers. For future research, one promising direction is to extend *TopER* to temporal graph learning tasks, enabling the capture of dynamic graph trajectories that reflect evolving user behaviors over time. Another avenue is the integration of *TopER* embeddings into graph foundation models, where the homogenization of graph structures could enhance the learning of transferable representations across different domains.

**Acknowledgments.** This work was partially supported by Canadian NSERC Discovery Grant RGPIN-2020-05665: Data Science on Blockchains, National Science Foundation under grants DMS-2220613, and DMS-2229417. The authors acknowledge the Texas Advanced Computing Center (TACC) at UT Austin for providing computational resources that have contributed to the research results reported within this paper. `http://www.tacc.utexas.edu`.

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

# Appendix

## A Further Experimental Details

### A.1 OGBG-MOLHIV Results

For the OGBG-MOLHIV dataset, we further evaluated the improvements of TopER with the addition of new filtration functions. Table 7 provides the performance of each TopER$-i$, where $i$ represents number of filtration functions used in the model, i.e., TopER-$i$ uses $\{(a_1, b_1, \ldots, a_i, b_i)\}$ as graph embedding where $(a_i, b_i)$ is the pivot and growth for function $f_i$. We used XG-Boost to rank the importance of filtration functions first, and the functions are added iteratively with this ranking. We fixed max-

Table 7: Results for OGBG-MOLHIV of each TopER$-i$.

| Method | Added Function | Valid. AUC | Test AUC |
|---|---|---|---|
| TopER-1 | degree-centrality | $72.76_{\pm 0.23}$ | $74.44_{\pm 0.20}$ |
| TopER-2 | atomic weight | $71.89_{\pm 0.12}$ | $74.25_{\pm 0.16}$ |
| TopER-3 | O. Ricci | $70.11_{\pm 0.28}$ | $76.79_{\pm 0.24}$ |
| TopER-4 | F. Ricci | $71.76_{\pm 0.18}$ | $78.15_{\pm 0.15}$ |
| TopER-5 | degree | $71.79_{\pm 0.35}$ | $79.26_{\pm 0.14}$ |
| TopER-6 | popularity | $72.27_{\pm 0.29}$ | $79.88_{\pm 0.24}$ |
| TopER-7 | closeness | $71.30_{\pm 0.18}$ | $\mathbf{80.21_{\pm 0.15}}$ |

imum tree depth $= 3$, learning rates $= 0.035$, subsample ratios $= 0.95$, the number of estimators $= 1000$, and the regularization parameter lambda $= 45$, where the objective function is rank:pairwise, with log loss as the evaluation metric. The seed is set to be $16$.

### A.2 Time Experiments for TopER vs. PH

To compare the time efficiency and performance of TopER and persistent homology (PH),

We conducted experiments using the same filtration function, the sublevel degree filtration. For PH, we applied Betti vectorization. Our results, summarized below, show that TopER is significantly faster than PH. Although both methods use the same filtration function, a key distinction lies in their embeddings: TopER generates 2D embeddings, whereas PH produces a vector with dimensionality equal to the number of thresholds in the filtration. Despite the considerable difference in dimensionality, TopER's performance with 2D embeddings remains comparable to that of PH.

Figure 5 shows that TopER scales efficiently with graph size, maintaining low runtime even with 100 filtration steps and high node degree. It processes graphs with up to 100,000 nodes in just over a minute, demonstrating its suitability for large-scale applications.

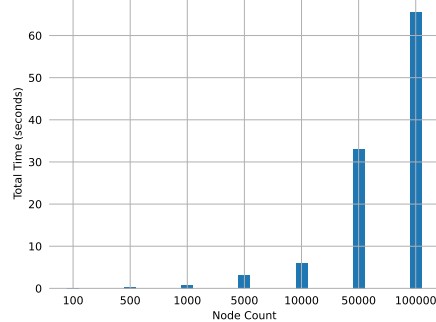

Figure 5: **Scalability.** TopER run time for synthetic power law graphs [HK02] with node degree filtration. The mean node degree is 30, and 100 filtration steps are used.

### A.3 Clustering Performances

In Table 9, we showcase our clustering performance across eight benchmark graph classification datasets using three widely adopted clustering metrics: Silhouette, Calinski-Harabasz, and Davies-Bouldin. These metrics serve as evaluative measures for assessing the efficacy of clustering algorithms

Table 8: Comparison of TopER-1 (only one filtration function) and PH in terms of time and accuracy across different datasets.

| Dataset | TopER-1 | | PH | | # Thresholds |
|---|---|---|---|---|---|
| | Time | Accuracy | Time | Accuracy | |
| BZR | 1.14 s | 82.73 ± 2.12 | 5.99 s | 83.70 ± 3.51 | 4 |
| IMDB-B | 3.27 s | 73.10 ± 4.18 | 319.95 s | 71.00 ± 4.07 | 65 |
| REDDIT-B | 107.65 s | 79.55 ± 2.20 | 9173.37 s | 84.50 ± 2.51 | 501 |

Table 9: The clustering performances of Spectral Embeddings and TopER with different metrics. Best performances are given in **blue**.

| | | | | Silhouette Scores ($\uparrow$) | | | | |
|---|---|---|---|---|---|---|---|---|
| **Method** | **BZR** | **COX2** | **MUTAG** | **PROT.** | **IMDB-B** | **IMDB-M** | **REDD-B** | **REDD-5K** |
| Spec Zoo | 0.050 | 0.049 | **0.344** | 0.050 | **0.097** | **-0.024** | 0.108 | -0.121 |
| degree | -0.108 | **0.414** | 0.258 | 0.048 | 0.030 | -0.032 | 0.049 | -0.169 |
| popularity | **0.249** | -0.015 | 0.134 | -0.000 | 0.008 | -0.159 | **0.196** | -0.173 |
| closeness | 0.019 | 0.036 | 0.036 | **0.086** | nan | nan | 0.087 | -0.185 |
| degree | 0.084 | 0.030 | 0.017 | 0.065 | 0.056 | -0.075 | 0.034 | **-0.067** |
| | | | | Calinski-Harabasz scores ($\uparrow$) | | | | |
| **Method** | **BZR** | **COX2** | **MUTAG** | **PROT.** | **IMDB-B** | **IMDB-M** | **REDD-B** | **REDD-5K** |
| Spec Zoo | 3.51 | 6.13 | **120.73** | 38.77 | **85.24** | **30.98** | 269.94 | 119.81 |
| degree | 0.42 | 1.06 | 11.29 | 130.07 | 60.52 | 3.92 | 97.85 | **1209.95** |
| popularity | 13.85 | **26.00** | 36.13 | 77.22 | 12.89 | 11.77 | **446.12** | 619.37 |
| closeness | **42.58** | 1.02 | 40.04 | 73.51 | 10.17 | 0.30 | 188.10 | 689.27 |
| F.Ricci | 4.92 | 0.48 | 11.82 | **151.64** | 11.68 | 1.03 | 92.14 | 454.34 |
| | | | | Davies-Bouldin scores ($\downarrow$) | | | | |
| **Method** | **BZR** | **COX2** | **MUTAG** | **PROT.** | **IMDB-B** | **IMDB-M** | **REDD-B** | **REDD-5K** |
| Spec Zoo | 7.25 | 6.07 | 0.95 | 4.55 | 2.78 | 10.73 | 2.20 | 25.74 |
| degree | 9.84 | **2.29** | **0.88** | 1.95 | 4.92 | 46.46 | 2.32 | 3.27 |
| popularity | 4.16 | 37.87 | 1.62 | 2.11 | 25.25 | **6.87** | **1.32** | 3.46 |
| closeness | **1.93** | 26.44 | 1.41 | 2.25 | 4.99 | 37.51 | 1.95 | **3.09** |
| F.Ricci | 4.19 | 7.20 | 1.27 | **1.54** | **2.19** | 10.35 | 1.83 | 5.41 |

in partitioning datasets into meaningful clusters. They gauge the degree of similarity or dissimilarity within and between clusters, offering insights into the quality of clustering outcomes. For precise definitions of Silhouette, Calinski-Harabasz, and Davies-Bouldin metrics, as well as additional details on clustering measures, refer to [GBC21].

## A.4 Number of Thresholds

In our experiments, we utilized a large number of thresholds to capture finer-grained information, as the model is computationally efficient and the additional cost of increasing the number of thresholds is minimal. Furthermore, in Table 10, we evaluated the model's performance with fewer thresholds and observed that it remains robust and highly effective even in such scenarios.

Table 10: The accuracy results of TopER with different numbers of thresholds.

| # Thresholds | PROTEINS | REDDIT-B | REDDIT-5K |
|---|---|---|---|
| 10 | $72.78_{\pm 4.04}$ | $90.55_{\pm 1.96}$ | $55.99_{\pm 1.97}$ |
| 20 | $74.31_{\pm 3.23}$ | $91.20_{\pm 1.66}$ | $55.91_{\pm 2.14}$ |
| 50 | $74.76_{\pm 4.55}$ | $92.05_{\pm 1.96}$ | $55.39_{\pm 2.10}$ |
| 100 | $73.85_{\pm 3.67}$ | $92.85_{\pm 1.18}$ | $55.51_{\pm 2.61}$ |
| 200 | $75.47_{\pm 3.06}$ | $93.15_{\pm 2.10}$ | $56.51_{\pm 2.04}$ |
| 500 | $74.58_{\pm 3.92}$ | $92.70_{\pm 2.38}$ | $56.51_{\pm 3.22}$ |

## A.5 Combining Filtration Functions

To assess the impact of embedding dimensions, we conducted new experiments evaluating the performance of the TopER model by progressively adding each filtration function step by step. This analysis provides insights into how the inclusion of additional filtration functions influences the model's performance. In Table 11, the TopER-n model represents the TopER utilizing n-filtration functions (2n features).

Table 11: Performance improvements achieved by integrating filtration functions into the TopER model. Here, TopER-n denotes the TopER model with n filtration functions.

| Dataset | TopER-1 | TopER-2 | TopER-3 | TopER-4 |
|---------|---------|---------|---------|---------|
| BZR | 82.48±1.98 | 84.70±2.84 | 85.66±5.00 | 86.68±3.81 |
| COX2 | 78.81±1.94 | 79.26±4.86 | 79.04±7.49 | 80.30±3.91 |
| MUTAG | 86.14±6.38 | 88.33±3.88 | 86.75±4.78 | 88.30±4.63 |
| PROTEINS | 74.03±2.71 | 74.67±2.73 | 75.21±3.39 | 75.65±3.87 |
| IMDB-B | 73.00±4.40 | 74.20±4.26 | 74.50±3.50 | 74.70±3.95 |
| IMDB-M | 48.73±4.33 | 49.80±2.94 | 49.73±4.18 | 49.87±4.00 |
| REDDIT-B | 81.95±2.74 | 90.45±2.55 | 91.05±2.62 | 91.50±2.01 |
| REDDIT-5K | 50.21±1.41 | 54.11±2.43 | 56.19±2.40 | 56.33±2.74 |

## A.6 TopER filtrations runtimes and substitute

**Filtration Timing Results.**   In  Table 12, we report the computation times (in seconds) for the TOPER filtration functions across various datasets. We also include the timings for the Heat Kernel Signature (HKS)  [SOG09], which can serve as an efficient substitute for the Ollivier–Ricci curvature in certain scenarios due to its faster computation while preserving relevant structural information about the graphs. The table below summarizes the observed computation times for each filtration type across multiple benchmark datasets.

Table 12: Computation times (in seconds) of different filtration functions across datasets.

| Filtration | BZR | COX2 | MUTAG | PROTEINS | IMDB-B | IMDB-M | REDDIT-B | REDDIT-5K |
|------------|-----|------|-------|----------|--------|--------|----------|-----------|
| Degree Centrality | 0.86 | 1.67 | 0.40 | 13.82 | 13.32 | 13.03 | 115.09 | 447.72 |
| Popularity | 0.73 | 1.36 | 0.58 | 5.81 | 13.21 | 15.60 | 111.11 | 414.04 |
| Closeness | 2.29 | 4.40 | 0.77 | 15.97 | 5.85 | 6.63 | 399.2 | 1274.00 |
| Forman Ricci | 0.80 | 1.56 | 0.58 | 4.63 | 8.29 | 7.38 | 258.32 | 693.18 |
| Ollivier Ricci | 130.04 | 121.42 | 48.98 | 313.21 | 277.66 | 428.32 | 1291.93 | 6640.95 |
| Degree | 1.14 | 1.16 | 0.61 | 2.62 | 3.27 | 4.22 | 107.65 | 363.32 |
| Weight | 2.25 | 2.80 | 0.90 | 12.89 | – | – | – | – |
| HKS | 2.07 | 1.94 | 0.56 | 16.90 | 15.54 | 15.72 | 470.01 | 1007.50 |

We evaluate the performance of our TopER model against state-of-the-art (SOTA) methods on benchmark graph classification datasets, including BZR, COX2, MUTAG, PROTEINS, IMDB-B, IMDB-M, REDDIT-B, and REDDIT-5K. Table 13 reports classification accuracy (mean ± standard deviation) across multiple runs. TopER generally achieves competitive results compared to SOTA. Ablation studies show the impact of Ricci curvature and Heat Kernel Signatures (HKS) on model performance. Notably, TopER without Ricci but with HKS recovers most of the performance lost when Ricci is removed, suggesting that HKS can serve as a viable replacement for Ollivier-Ricci curvature in capturing structural information.

Table 13: Graph classification accuracy (mean ± std) for different models across benchmark datasets. Highest scores per dataset are **bold blue**, second-highest are underlined blue.

| Model | BZR | COX2 | MUTAG | PROTEINS | IMDB-B | IMDB-M | REDDIT-B | REDDIT-5K |
|-------|-----|------|-------|----------|--------|--------|----------|-----------|
| SOTA | 89.00±5.00 | **85.53±1.60** | **92.63±2.58** | **77.30±0.89** | **75.08±0.31** | **52.81±0.31** | 91.03±0.22 | 56.75±0.18 |
| TopER | **90.13±4.14** | 82.01±4.59 | 90.99±6.64 | 74.58±3.92 | 73.20±3.43 | 50.00±4.02 | 92.70±2.38 | 56.51±2.22 |
| TopER w/o O. R. | 87.00±4.30 | 77.96±8.38 | 87.78±7.84 | 74.04±3.86 | 73.50±3.53 | 50.00±5.44 | 91.90±2.63 | 56.37±1.89 |
| TopER w/o O. R.& w/ HKS | 89.63±3.65 | 81.58±3.54 | 92.08±4.23 | 75.20±3.59 | 75.00±3.49 | 50.67±5.58 | **92.75±2.47** | **57.33±2.02** |

## A.7 Hyperparameters

Our proposed MLP algorithm is constructed with a single hidden layer. The output layer's activation function is set to log softmax, and the loss function we used is Negative Log Likelihood Loss. The learning rate is chosen between 0.01 and 0.001. Subsequently, we investigate the impact of the number of neurons in the hidden layer, considering values from the set {16, 64, 128}. The optimizer is set to be Adam, and the number of epochs is 500. To prevent large weights and overfitting, we apply L2 regularization coefficients of 1e-3, 1e-4. The activation function for the hidden layer varies between ReLU, GeLU, and ELU. Lastly, we consider the cases of adding or not a batch normalization

---

**Algorithm 1** TopER: Topological Evolution Rate

---

**Input:** Graph $\mathcal{G}$, Filtration function $f : \mathcal{V} \to \mathbb{R}$, Threshold set $\mathcal{I} = \{\epsilon_i\}_{i=0}^{n}$
**Output:** TopER vector $\mathcal{T}_f(\mathcal{G}, \mathcal{I})$
Initialize lists $\mathcal{X} = []$, $\mathcal{Y} = []$
**for** $i = 1$ **to** $n$ **do**
   $\mathcal{G}_i \leftarrow$ Induced subgraph of $\mathcal{G}$ where $\mathcal{V}_i \subseteq f^{-1}([\epsilon_0, \epsilon_i])$
   $x_i \leftarrow |\mathcal{V}_i|$
   $y_i \leftarrow |\mathcal{E}_i|$
   Append $x_i$ to $\mathcal{X}$
   Append $y_i$ to $\mathcal{Y}$
**end for**
Fit a line $\mathcal{L}(x) = a + bx$ to pairs $(x_i, y_i)$ from $\mathcal{X}$ and $\mathcal{Y}$ using least squares
Extract coefficients $a$ and $b$
**Return** $(a, b)$ as the TopER vector $\mathcal{T}_f(\mathcal{G}, \mathcal{I})$

---

layer to the output of the hidden layer and setting dropout values to be 0.0 or 0.5. In Table 14, we provide the details for each dataset. The last column shows the number of TopER features used for each dataset after the feature selection step.

Table 14: Employed hyperparameters for each dataset.

| Dataset | Neurons | Dropout | Batch Norm. | Decay | Learning rate | Activation | TopER Dim. |
|---|---|---|---|---|---|---|---|
| **BZR** | 64 | 0.5 | True | 1e-4 | 0.001 | gelu | 26 |
| **COX2** | 128 | 0 | True | 1e-4 | 0.01 | relu | 26 |
| **MUTAG** | 16 | 0.5 | False | 1e-3 | 0.01 | gelu | 20 |
| **PROTEINS** | 64 | 0.5 | True | 1e-3 | 0.01 | elu | 26 |
| **IMDB-B** | 128 | 0 | False | 1e-3 | 0.001 | relu | 20 |
| **IMDB-M** | 16 | 0 | False | 1e-3 | 0.01 | elu | 20 |
| **REDDIT-B** | 64 | 0.5 | False | 1e-3 | 0.01 | relu | 24 |
| **REDDIT-5K** | 128 | 0 | False | 1e-3 | 0.01 | elu | 14 |

## B More on TopER

### B.1 Refining the point set

While we have described the main steps of TopER in Section 3, due to the repetitions of the points in $\mathcal{A} = \{(x_i, y_i)\} \subset \mathbb{R}^2$, there are some choices to be made before defining the set $\mathcal{A}$ (i.e., $\mathcal{X}$ and $\mathcal{Y}$) to get the best fitting function $L : \mathcal{X} \to \mathcal{Y}$. The main reason is that the set $\{(x_i, y_i)\}_{i=1}^{N}$ can contain repetitions of $x$-values ($x_i = x_{i+1}$), repetitions of $y$-values ($y_i = y_{i+1}$) or repetitions of both ($(x_i, y_i) = (x_{i+1}, y_{i+1})$) depending on the filtration function, the threshold set $\mathcal{I}$, and the graph $\mathcal{G}$.

For the filtrations induced by *node filtration functions*, the number of edges can not change unless the number of nodes changes, i.e., $x_i = x_{i+1} \Rightarrow y_i = y_{i+1}$. Hence, with this elimination, we still allow keeping $y$-values the same while $x$-values are increasing. This means there can be horizontal jumps in $\mathcal{A}_u$. In this paper, to eliminate all horizontal jumps for filtrations with node functions, we eliminate all repetitions of $y$-values from $\mathcal{A}_u$. In particular, we remove all the points with the same $\widehat{y}$-value and add a point with a mean of $x$-values. In other words, if $y_i = y_{i+1} = \cdots = y_{i+k} = \widehat{y}$, we define $\widehat{x} = \text{mean}\{x_i, x_{i+1}, \ldots, x_{i+k}\}$. Then, we replace (k+1) points $\{(x_i, \widehat{y}), (x_{i+1}, \widehat{y}), \ldots, (x_{i+k}, \widehat{y})\}$ with one point $(\widehat{x}, \widehat{y})$ in $\mathcal{A}_u$. This process eliminates all repetitions and horizontal jumps in $\mathcal{A}$, and we define our best-fitting line on this refined set.

### B.2 TopER with Alternative Quantities

While we use the most general quantities for a graph—the count of vertices and edges—in our algorithm, depending on the problem, there might be other induced quantities $(x_i, y_i)$ for a given subgraph $\mathcal{G}_i$ which can give better vectors. To keep the line-fitting approach meaningful in our model, as long as the sequences $\{x_i\}$ and $\{y_i\}$ are monotone like our node-edge counts above, for a given

dataset in a domain (e.g., biochemistry, finance), one can use other domain-related quantities induced by substructure $\mathcal{G}_i$ as a $(x_i, y_i)$ pair to obtain a TopER vector.

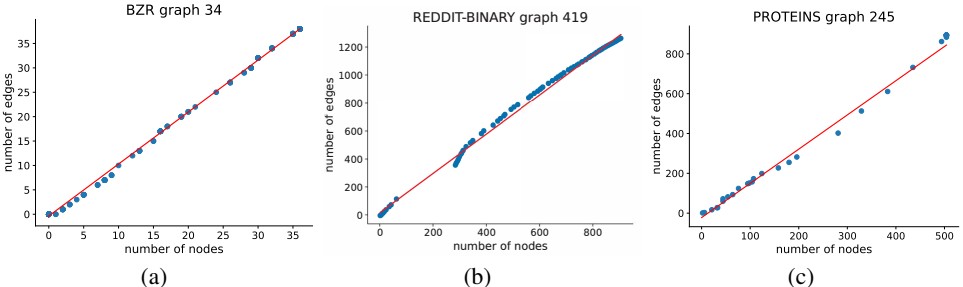

Figure 6: **Linear Fit.** TopER summarizes the growth behavior in the graph induced by filtration with a linear fit.

## B.3  Linear or Higher Order Fitting

In our experiments, we observe that linear fitting captures the growth information for node-edge pair $\{(x_i, y_i)\}$ well (See Figure 6), and quadratic fit and linear fit stay very close to each other. However, if one decides to use other quantities as described above and loses the monotonicity of the sequences $\{x_i\}$ and $\{y_i\}$, trying higher order fits (e.g., $y = ax^2 + bx + c$) can be more meaningful. In Table 15, we present the average of the coefficients of quadratic terms when we use quadratic fit for the datasets, i.e., if we fit $y = a + bx + cx^2$ polynomial, we observe that the quadratic term $cx^2$ is mostly negligible, and tends to be a linear fit.

Table 15: Average of $x^2$ coefficient across datasets for quadratic fitting.

| Dataset | BZR | COX2 | MUTAG | REDDIT-5k |
|---|---|---|---|---|
| **Average of $x^2$ Coefficient** | $4.71 \times 10^{-5}$ | $6.61 \times 10^{-4}$ | $1.16 \times 10^{-2}$ | $1.78 \times 10^{-5}$ |

## B.4  Interpreting TopER

Our approach involves accurately modeling the evolution of a graph throughout the filtration process. One can easily identify clusters for each class and outliers in the other datasets given in Figure 1(a) and make inferences about the different clusters and outliers. Furthermore, when the pivot $a_f$ is positive or negative, it can be interpreted as graph density behavior in the filtration sequence (See Figure 7).

## C  Proofs of Stability Theorems

In this part, we prove the stability results for our TopER.

**Lemma 3.6**. [ST20] Let $\mathcal{X}$ be a compact metric space, and $f, g : \mathcal{X} \to \mathbb{R}$ be two filtration functions. Then, for any $p \geq 1$, we have $\mathcal{W}_p(\mathrm{PD}_k(\mathcal{X}, f), \mathrm{PD}_k(\mathcal{X}, g)) \leq \|f - g\|_p$

The next lemma is on the stability of Betti curves by [DG23] [Proposition 1].

**Lemma 3.7**. [DG23] Let $\beta_k(\mathcal{X})$ is the $k^{th}$ Betti function obtained from the persistence module $\mathrm{PM}_k(\mathcal{X})$.
$$\|\beta_k(\mathcal{X}) - \beta_k(\mathcal{Y})\|_1 \leq 2\mathcal{W}_1(\mathrm{PD}_k(\mathcal{X}), \mathrm{PD}_k(\mathcal{Y}))$$

Now, we are ready to prove our stability result.

**Theorem 3.4** Let $\mathcal{X}$ be a compact metric space, and $f, g : \mathcal{X} \to \mathbb{R}$ be two filtration functions. Then, for some $C > 0$,
$$\|\mathrm{TE}_f(\mathcal{X}) - \mathrm{TE}_g(\mathcal{X})\|_1 \leq \mathrm{C} \cdot \mathcal{W}_1(\mathrm{PD}_k(\mathcal{X}, f), \mathrm{PD}_k(\mathcal{X}, g))$$

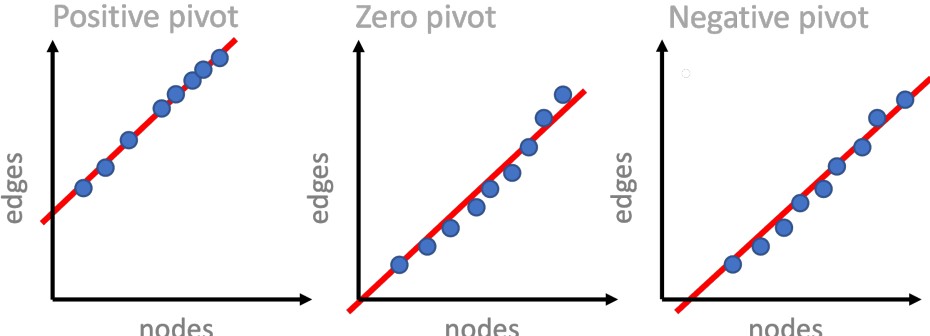

Figure 7: **Pivot Behavior.** A graph can exhibit three distinct pivot behaviors. Positive pivot graphs display a cluster of vertices that are closely interconnected and appear early in the filtration process. On the other hand, negative pivot graphs feature loosely connected nodes where the edges enter the filtration at a later stage. Graphs with zero pivot are usually quasi-complete graphs.

*Proof.* We will utilize the stability theorems from topological data analysis given above.

First, we employ the stability of Betti curves by Lemma 3.7.

$$\|\beta_k(\mathcal{X}) - \beta_k(\mathcal{Y})\|_1 \leq 2\mathcal{W}_1(\mathrm{PD}_k(\mathcal{X}), \mathrm{PD}_k(\mathcal{Y})) \tag{1}$$

Hence to obtain $\mathrm{TE}_f(\mathcal{X}) = (a_f, b_f)$, we fit least squares line $y = a_f + b_f x$ to the set of $N$ points in $\mathbb{R}^2$, i.e., $\mathcal{Z}_f = \{(\beta_0^f(\epsilon_i), \beta_1^f(\epsilon_i))\}_{i=1}^N$. Similarly, we obtain $\mathrm{TE}_g(\mathcal{X}) = (a_g, b_g)$ by fitting least squares line to $\mathcal{Z}_g = \{(\beta_0^g(\epsilon_i), \beta_1^g(\epsilon_i))\}_{i=1}^N$. By Equation (1), we have

$$\mathbf{D}_H(\mathcal{Z}_f, \mathcal{Z}_g) \leq 4\mathcal{W}_1(\mathrm{PD}_k(\mathcal{X}), \mathrm{PD}_k(\mathcal{Y})) \tag{2}$$

where $\mathbf{D}_H(\mathcal{Z}_f, \mathcal{Z}_g)$ represent Hausdorff distance between the point clouds $\mathcal{Z}_f$ and $\mathcal{Z}_g$ in $\mathbb{R}^2$.

Now, by the stability of least squares fit with respect to Hausdorff distance ([CHM12] [Theorem 3.1]), we have

$$\|\mathrm{TE}_f(\mathcal{X}) - \mathrm{TE}_g(\mathcal{X})\|_1 \leq \mathrm{C} \cdot \mathbf{D}_H(\mathcal{Z}_f, \mathcal{Z}_g) \tag{3}$$

Hence, when we combine Equations (2) and (3), we have

$$\|\mathrm{TE}_f(\mathcal{X}) - \mathrm{TE}_g(\mathcal{X})\|_1 \leq \mathrm{C} \cdot \mathcal{W}_1(\mathrm{PD}_k(\mathcal{X}), \mathrm{PD}_k(\mathcal{Y}))$$

The proof follows. □

By combining the above result with Lemma 3.6, we obtain the following corollary.

**Corollary 3.5** Let $\mathcal{X}$ be a compact metric space, and $f, g : \mathcal{X} \to \mathbb{R}$ be two filtration functions. Then, for some $C > 0$,

$$\|\mathrm{TE}_f(\mathcal{X}) - \mathrm{TE}_g(\mathcal{X})\|_1 \leq \mathrm{C} \cdot \|f - g\|_1$$

*Proof.* By Lemma 3.6, we have

$$\mathcal{W}_1(\mathrm{PD}_k(\mathcal{X}, f), \mathrm{PD}_k(\mathcal{X}, g)) \leq \|f - g\|_1 \tag{4}$$

By Theorem 3.4, we have

$$\|\mathrm{TE}_f(\mathcal{X}) - \mathrm{TE}_g(\mathcal{X})\|_1 \leq \mathrm{C} \cdot \mathcal{W}_1(\mathrm{PD}_k(\mathcal{X}, f), \mathrm{PD}_k(\mathcal{X}, g)) \tag{5}$$

By combining Equations (4) and (5), we conclude

$$\|\mathrm{TE}_f(\mathcal{X}) - \mathrm{TE}_g(\mathcal{X})\|_1 \leq \widehat{\mathrm{C}} \cdot \|f - g\|_1$$

The proof follows. □

# D    Synthetic Experiments

This section describes experiments performed on the Erdos-Renyi synthetic graph. We conducted two experiments by applying TopER on this graph and Principal Component Analysis (PCA). The goal is to compare the performance, runtime, and interpretability of the two models. From the plots shown in Figures 8, 9, and 10, we see that TopER is interpretable compared to when PCA is used.

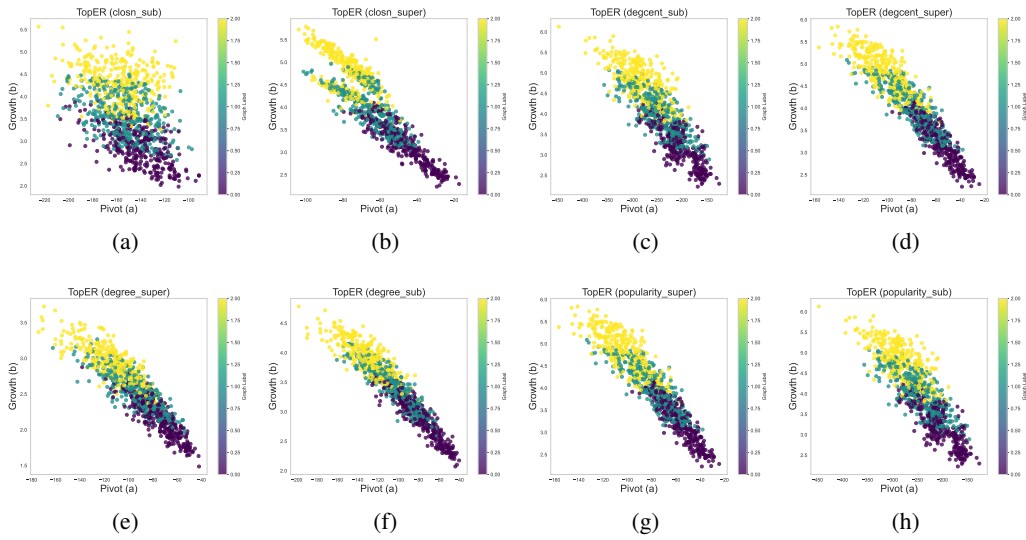

Figure 8: TopER plots showing pivot vs growth for each function—degree centrality, closeness, degree, and popularity when threshold=50.

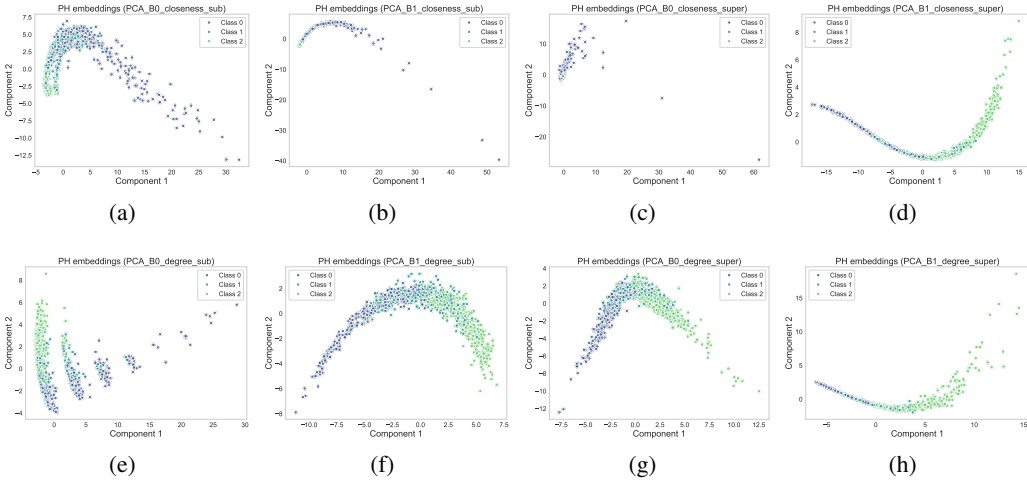

Figure 9: Persistent Homology PCA plots embeddings (Betti 0 and Betti 1) showing component 1 vs component 2 for each function—closeness and degree when threshold=50 and the respective filtrations —sublevel and superlevel.

# E    Broader Impact

This work advances the field of graph representation learning by introducing a topological approach that is both interpretable and scalable. By leveraging the structural insights of persistent homology without incurring its prohibitive computational costs, *TopER* enables more efficient and insightful analysis of complex graph-structured data. This has the potential to benefit a range of scientific and

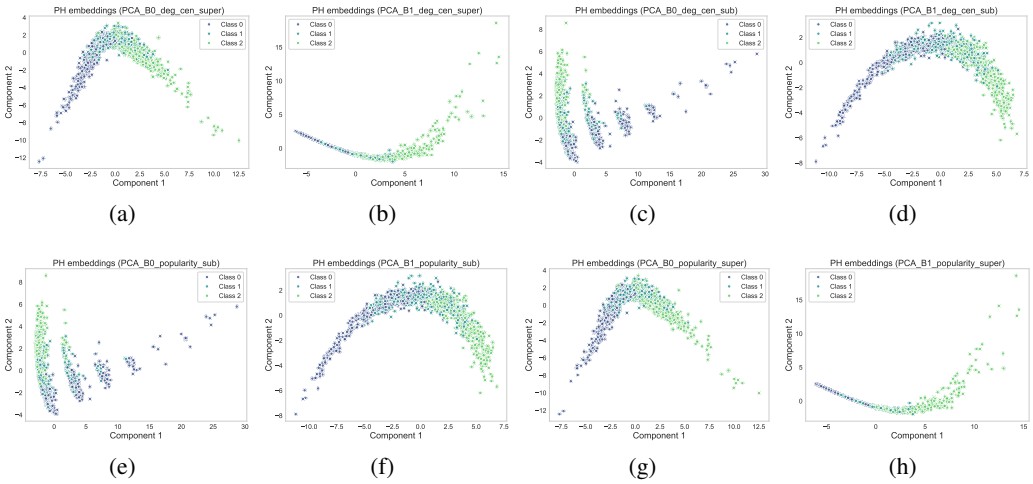

(a)  (b)  (c)  (d)

(e)  (f)  (g)  (h)

Figure 10: Persistent Homology PCA plots embeddings (Betti 0 and Betti 1) showing component 1 vs component 2 for each function—degree centrality and popularity when threshold=50 and the respective filtrations —sublevel and superlevel.

industrial domains where graph data is prevalent, including bioinformatics, social network analysis, and infrastructure monitoring. In particular, the ability to generate low-dimensional and interpretable embeddings could assist researchers in visual analytics, pattern discovery, and model debugging. At the same time, we acknowledge that the use of graph representations—especially in social networks or biological datasets—may carry ethical concerns around data privacy, representational bias, or unintended consequences of automated decision-making. While *TopER* itself is an unsupervised and domain-agnostic method, its application must be governed by domain-specific ethical considerations. To support responsible use, we emphasize interpretability and transparency in our design, and we release our code and visualizations to promote reproducibility and community oversight.

