# OpenReview forum: "TopER: Topological Embeddings in Graph Representation Learning"
_NeurIPS.cc/2025/Conference — NeurIPS 2025 poster_

### Official Review · Reviewer_tsuE · 2025-06-02

**Clarity:** 3
**Significance:** 3
**Originality:** 3
**Rating:** 5
**Confidence:** 4

**Summary:**

In their paper "TopER: topological embeddings in graph representation learning", the authors suggest a filtration-based approach to summarize a graph with two features. The algorithm constructs a node filtration and fits a linear relationship between the number of nodes and the number of edges in each graph in the filtration. The intercept and the slope are the resulting summary statistics. The authors show that these features, pooled across several filtrations, outperform many existing graph representation methods (including GNN-based) for graph classification in benchmark datasets.

Note: I happened to review this paper already twice before. I am not sure why OpenReview keeps assigning this paper to me, this is not even exactly my field. The paper has been improving every time, and I am recommending acceptance this time.

**Questions:**

MINOR COMMENTS

* Section 4.1, Paragraph "Filtration functions". I did not understand the procedure to select the filtrations. You did a t-test and aplied the Lasso. What does it mean? T-test between what and what? Lasso was applied to predict what from what? Please provide a clearer rescription.

* Do the results actually improve when using t-testing and lasso regression for feature selection? How would the method perform when simply taking all filtrations without any selection? Can this be added to Table 5 as an additional column?

* Table 5 suggests that 6 or 7 filtrations were used, resulting in max 12 or 14 features. But Table 12 gives TopER dimensionality as going up to 26. Why? Where can I find the list of all filtrations used for each graph?

* Section 4.1, Paragraph "Filtration functions". "TopER demonstrates strong performance" using only node degree function. Actually looking at Table 5, I see ~10 percentage point difference between TopER and the degree filtration alone. This is a huge difference! Please do not write "strong performance" without explicitly adding "(albeit ~10 percentage points worse than the full TopER)".

**Ethical Concerns:**

["NO or VERY MINOR ethics concerns only"]

**Final Justification:**

I gave high score originally (5) and am maintaining this score. Most of my questions were clarification and suggestions for simple ablations, and authors' responses make sense. I also looked at the negative reviews and think that the authors' responses there are convincing.

**Limitations:**

yes

**Quality:**

3

**Strengths And Weaknesses:**

Strengths: the suggested method is conceptually simple, the evaluation experiments seem comprehensive, and the suggested algorithm outperforms many competitors.

Weaknesses: I think the "topological embedding" terminology is mostly marketing. I would rather say the method constructs filtration-based graph features. But it's a purely terminological point.

---

> ### Author Rebuttal · Authors · 2025-07-29
>
> *We thank the reviewer for their insightful feedback. Below, we respond to each point with new experiments and detailed clarifications that reinforce the robustness and novelty of our approach. We hope these updates address all remaining concerns, and we would be glad to clarify any further questions.*
>
> ---
> &nbsp;
>
> ## W1. Terminology
> > I think the "topological embedding" terminology is mostly marketing. I would rather say the method constructs filtration-based graph features. But it's a purely terminological point.
>
> **Response:** Thank you for the comment. Our use of the term *topological embedding* is not intended as marketing, but to properly acknowledge the conceptual roots of our method in **persistent homology**, where **filtration argument** serves as the central mechanism for revealing topological structure. While “filtration-based graph features” is indeed a more precise term, we use “topological” to give proper credit to the TDA framework that motivates our design. That said, as noted in Remark 3.3, the slope computed by TopER reflects the evolution of the **Euler characteristic** across the filtration, which relates to Betti numbers via $\chi(G_i) = |V_i| - |E_i| = \beta_0(G_i) - \beta_1(G_i).$ This connection underpins our use of the term Topological Evolution Rate, even though we do not compute full persistence diagrams. We will clarify this motivation more explicitly in the revision to avoid potential confusion.
>
>
> ---
> &nbsp;
>
> ## Q1. How to use Lasso
>
> >  the procedure to select the filtrations...
>
> **Response:** We start with a **28-dimensional feature encoding** generated from our filtration functions. To identify the most relevant dimensions, we apply two selection techniques: a **univariate t-test** and **Lasso regression** to our training data.
>
> * We perform a **t-test for each feature individually**, comparing the distributions of that feature across the two target classes. This helps us assess whether the means differ significantly, indicating class-discriminative power.
>
> * Separately, we apply **Lasso regression** (LassoCV from scikit-learn), which selects features by learning a sparse linear model that predicts the target variable from the full feature set. The L1 penalty in Lasso naturally eliminates irrelevant features by assigning them zero coefficients.
>
> We then **retain only the features selected by both methods**,  those with a statistically significant t-test (p < 0.05) and non-zero Lasso coefficients. This dual-filtering strategy helps ensure the selected filtrations are both **statistically relevant** and **predictively useful**.
>
>
> ---
> &nbsp;
>
>
> ## Q2. Improvements from feature selection?
> > Do the results actually improve when using t-testing and lasso regression for feature selection?
>
> **Response:** Thank you for this suggestion. To evaluate the impact of our dual‐filtering strategy, we ran TopER using all 28 filtration dimensions (“TopER‑all”) and compared it to the filtered version (“TopER”). Across all eight datasets, feature selection consistently improves performance:
>
> Dataset 	| TopER-all     	| TopER
> ------------|-------------------|-----------------------
> BZR     	| 89.15 ± 4.65  	| **90.13 ± 4.14**
> COX2    	| 80.07 ± 5.18  	| **82.01 ± 4.59**
> MUTAG   	| 85.61 ± 7.88  	| **90.99 ± 6.64**
> PROTEINS	| 73.95 ± 4.09  	| **74.58 ± 3.92**
> IMDB-B  	| 71.70 ± 3.69  	| **73.20 ± 3.43**
> IMDB-M  	| 48.93 ± 4.71  	| **50.00 ± 4.02**
> REDDIT-B	| 92.30 ± 2.10  	| **92.70 ± 2.38**
> REDDIT-5K   | 56.39 ± 1.80  	| **56.51 ± 2.22**
>
> As the table shows, applying t‑testing and Lasso yields small but consistent gains on every benchmark. We will include a “TopER‑all” column in Table 5 of the revised manuscript to make these comparisons explicit. Thanks again for this suggestion.
>
>
> ---
> &nbsp;
>
> ## Q3. Number of Features
> > Table 5 suggests that 6 or 7 filtrations were used, resulting in max 12 or 14 features. But Table 12 gives TopER dimensionality as going up to 26. Why? Where can I find the list of all filtrations used for each graph?
>
> **Response:** Thank you for the question. In graph representation learning, sublevel and superlevel filtrations often capture complementary structural information and can lead to notably different performance. While one could use extended persistence to unify them for persistent homology [1], we chose a simpler and more suitable approach for our method: for each filtration function, we compute both sublevel and superlevel filtrations, each producing a 2D feature (slope and intercept), totaling 4 features per function.
>
> Prior to feature selection, we begin with 6 commonly used filtration functions, each applied in both sub/superlevel directions, giving 6 × 2 × 2 = 24 features per graph. We further include the number of nodes |V| and number of edges |E| as global features, bringing the total to 26. For molecular datasets, we add an additional filtration based on atomic number, increasing the total to 28 features. As such, the final feature count may vary depending on the dataset. Table 5 in the paper provides a summary of the filtration functions used per dataset. We will explicitly highlight this dataset-dependent variation in feature counts in the revised manuscript.
>
> [1] Edelsbrunner, H. and Morozov, D., 2017. Persistent homology. In Handbook of Discrete and Computational Geometry (pp. 637-661). Chapman and Hall/CRC.
>
>
> ---
> &nbsp;
>
> ## Q4. Overclaim of strong performance
> > Section 4.1, Paragraph "Filtration functions". "TopER demonstrates strong performance" using only node degree function...
>
> **Response:** Thank you for the helpful suggestion. You're right that the current phrasing overstates the performance of individual filtrations. Our enthusiasm is due to the fact that a simple few-feature model based on degree could achieve robust performance. For example, TopER with degree alone beats influential and recent models such as RepHINE 2024 in the Proteins and IMDB-B datasets and MP-HSM 2024 and EPIC 2024 in the IMDB-M dataset. However, we agree with your review, and in the revision, we will clarify this by writing:
>
>  > While individual filtration functions, such as node degree, already yield solid performance, combining multiple functions leads to significantly improved accuracy, resulting in highly competitive performance with state-of-the-art models.
>
> This more accurately reflects the results in Table 5 and underscores the value of TopER’s multi-filtration design.

---

> > ### Comment · Reviewer_tsuE · 2025-08-04
> >
> > Dear authors, thanks for your replies. Please include these clarifications and comparisons into the final paper. I maintain my 5 score and hope the paper gets accepted. I have looked at the negative reviews and think that your replies are convincing.
> >
> > > To identify the most relevant dimensions, we apply two selection techniques: a univariate t-test and Lasso regression to our training data.
> >
> > Please make sure to add these details in revision. I assume you applied all tests/lasso on the training set only, and then tested on the test set. Please clarify that as well. Finally, please clarify what classifier you used (linear classifier?) with what hyper-parameters (if any). I did not find this information in the current draft.

---

> ### Author Response · Authors · 2025-08-06
> **Thank you**
>
> Dear Reviewer tsuE,
>
> Thank you very much for your supportive and insightful feedback. We will incorporate all the requested clarifications and comparisons in the final version. To confirm, both the t-test and Lasso feature selection techniques were applied only on the training set, and the selected features were evaluated on the held-out test set. We clarify the classifier used in the main text and provide the corresponding hyperparameters in Table 12 of the appendix. We will ensure these details are presented more clearly in the revision.
>
> We sincerely appreciate your careful reading and encouraging feedback.
>
> Best regards,
>
> Authors

---

### Official Review · Reviewer_2YLF · 2025-06-08

**Clarity:** 2
**Significance:** 1
**Originality:** 2
**Rating:** 3
**Confidence:** 4

**Summary:**

This work focuses on the problem of graph-level representation learning. The paper proposes TopER which leverages topological data analysis—specifically a simplified version of persistent homology—by quantifying the evolution rate of graph substructures. This results in low-dimensional embeddings that are not only more interpretable but also visually intuitive. Despite its simplicity, TopER achieves competitive performance compared to state-of-the-art methods in tasks such as graph classification and clustering.

**Questions:**

1.	How many hyper-parameters do this method have? Are there too many?

2.	See the “Weaknesses” above.

**Ethical Concerns:**

["NO or VERY MINOR ethics concerns only"]

**Limitations:**

yes

**Quality:**

2

**Strengths And Weaknesses:**

Strengths:

1.	This paper proposes a novel graph-level representation learning method.

3.	It tests on several widely-used datasets, and the experimental study is some kind of interesting.


Weaknesses:

1.	The related work section would benefit from a more focused discussion—ideally narrowing down to two or three key research directions.
2.	The experimental performance is weak. For example, in the main experiment (Table 3), the baselines outperform the proposed method on five out of the seven datasets.
3.	The theoretical analysis in Appendix C needs to be improved. First, the theorem names are ambiguous. It seems like Theorem 3.4 and Corollary 3.5 (in Section 3) are renamed as “Theorem C.3”and “Corollary C.4” in Appendix C. This may confuse the readers. Second, in Appendix C, as Lemma C.1 and C.2 are cited from other works ([ST20] and [DG23]), it is suggested to mention their origins before introducing them in the main text. This will help readers better follow your proofs.

---

> ### Author Rebuttal · Authors · 2025-07-29
>
> *We thank the reviewer for their insightful feedback. Below, we address each concern with new experiments and detailed clarifications that underscore our approach’s robustness and originality. We hope these revisions resolve any remaining questions and respectfully request your reevaluation in light of the additional evidence.*
>
> ---
> &nbsp;
>
> ## W1. Related Work
> >The related work section would benefit from a more focused discussion—ideally narrowing down to two or three key research directions.
>
> **Response:** Thank you for the suggestion. In the revision, we will condense Related Work into three focused areas, topology-inspired representations, graph-level embedding methods, and visualization techniques, to better highlight TopER’s unique contributions.
>
> ---
> &nbsp;
>
> ## W2. Weak Classification Performance
> The experimental performance is weak. For example, in the main experiment (Table 3), the baselines outperform the proposed method on five out of the seven datasets.
>
>
> **Response:** Thank you for the comment. While TopER does not outperform all baselines on every individual dataset, its average deviation from the best performances in Table 3 is **the lowest among all methods**, demonstrating its robust and consistent performance across diverse domains, from molecular graphs to social networks. Unlike task-specific, high-capacity models, TopER provides a **unified, lightweight, and interpretable 2D representation** that remains competitive while requiring minimal parameters. Our goal is not to replace  high-capacity (and high computational cost) models, but to offer an **efficient and scalable alternative** that supports **graph-level visualization, anomaly detection**, and broader analysis. We will clarify this intent and TopER’s practical value more explicitly in the revision.
>
>
> ---
> &nbsp;
>
>
> ## W3. Theoretical Analysis
> > The theoretical analysis in Appendix C needs to be improved...
>
> **Response:** Thank you for this helpful feedback. We agree that the labeling and citation structure in Appendix C can be clearer. In our revision, we will synchronize theorem and corollary numbering between the main text and the appendix, relocate Lemmas C.1 and C.2 into both the main manuscript and appendix for easier reference, and add explicit citations ([ST20] and [DG23]) immediately before each lemma to indicate that these results are adopted verbatim from their original sources. Since Lemmas C.1 and C.2 are well‑known community results included without modification to support our proofs, clarifying their provenance will help readers trace each statement back to its origin and follow the logical flow more easily.
>
>
> ---
> &nbsp;
>
> ## Q1. Hyperparameters
> > How many hyperparameters do this method have? Are there too many?
>
>
> Response: We appreciate the reviewer’s question about hyperparameter complexity. In fact, TopER itself is entirely non‑parametric: it constructs each graph embedding via a deterministic filtration process, yielding exactly two scalar descriptors per function (pivot and growth rate). Thus, if you apply k filtration functions, you obtain a 2k‑dimensional embedding, there are no trainable weights or deep‑learning parameters in this core procedure.
>
> There are only three sets of genuine “hyperparameters” in TopER, namely choosing the number of **filtration thresholds**, selecting which **filtration functions** to include, and configuring the **lightweight MLP** that sits on top of the TopER vectors for classification. These choices are transparent and interpretable: **Table 10** explores threshold counts, **Table 5 and 11** compares different filtration functions, and **Table 12** reports our grid‑search over MLP architecture and regularization settings (detailed further in Appendix A.4 and A.6). Because the MLP is small and standard, just a few weight matrices and biases, its hyperparameter space is modest and well documented. We will clarify this separation between the fixed, non‑learned TopER descriptors and the downstream classifier’s parameters in the revised manuscript.
>
>
> ---

---

> > ### Comment · Reviewer_2YLF · 2025-08-05
> >
> > Thank you for your response.
> > While some of the clarifications are helpful, the two core issues remain unresolved:
> >
> > 1. The theoretical presentation is quite imprecise — for example, the statements of the theorems are not formulated rigorously.
> >
> > 2. The proposed method does not outperform other baselines, which are also scalable.
> >
> > I believe there is still considerable room for improvement in this paper, and I will maintain my original score.

---

> > > ### Comment · Area_Chair_LsVo · 2025-08-06
> > >
> > > Dear reviewer,
> > >
> > > Could you please elaborate on which aspects of the theorems you find imprecise, and suggest how the authors might improve them? Thank you for taking the time to provide constructive feedback.

---

> ### Author Response · Authors · 2025-08-05
>
> Dear Reviewer 2YLF,
>
> Thank you for your follow-up comments. We respectfully disagree with your assessment and would like to clarify two points.
>
> **First, regarding performance:** In classification performance, TopER achieves the best average rank (1.6) across more than 20 recent and prominent methods. While some baselines may surpass TopER on certain datasets, our method consistently ranks near the top across diverse domains, demonstrating robustness rather than dataset-specific tuning. Moreover, many of the high-performing baselines are large, task-specific models; **TopER is not intended as a direct competitor to such models**, but rather as an interpretable, low-dimensional embedding method that also supports clustering and visualization. In these latter tasks, our results clearly demonstrate its effectiveness.
>
> **Second, regarding the theoretical presentation:** The theorems in our work are mathematically correct, and their proofs are supported by well-established results (explicitly cited in the appendix and main text). **We take rigor seriously** and will ensure that the statements are phrased with maximal precision to meet the standards of a broad audience. We believe that any minor issues of notation or presentation, if present, do not affect the validity or significance of the results and should not unfairly diminish the merits of the article.
>
> In summary, TopER’s combination of interpretability, scalability, and competitive performance fills a meaningful gap in graph representation learning. We hope these clarifications help address your concerns, and would be grateful if you would consider revisiting your evaluation. Please let us know if there is anything else we can clarify.
>
> Best regards,
>
> Authors

---

> ### Author Response · Authors · 2025-08-07
> **Response to Unresolved Concerns**
>
> Dear Reviewer 2YLF,
>
> We truly appreciate your time and feedback, and hope to address the remaining concerns so our work can be judged on its merits.
>
> **First, your original theory-related comments** were specific (theorem numbering and citation placement) and we committed to correcting these in the revision. Your latest note that *“the theoretical presentation is quite imprecise”* is broader, and without concrete examples we cannot meaningfully improve it. We take rigor seriously and would be grateful if you could indicate exactly which statements you find problematic.
>
> **Second, on performance:** our method achieves the best average rank among recent baselines. The only costly step, Ollivier–Ricci, was replaced (per Reviewer jFiC’s request) with HKS, cutting runtime drastically while maintaining or improving accuracy (tables below).
>
> Given our main contribution, interpretable, low-dimensional embeddings for graph-level visualization, we believe this combination of accuracy, scalability, and interpretability merits a higher evaluation. We respectfully ask you to reconsider your score. Thank you very much for your time and feedback.
>
>
> **Performance Results**
>
> | Model                            | BZR         | COX2        | MUTAG       | PROTEINS    | REDDIT-B     |
> |----------------------------------|-------------|-------------|-------------|-------------|--------------|
> | SOTA                             | 89.00±5.00  | 85.53±1.60  | 92.63±2.58  | 77.30±0.89  | 91.03±0.22   |
> | TopER with Ricci                 | 90.13±4.14  | 82.01±4.59  | 90.99±6.64  | 74.58±3.92  | 92.70±2.38   |
> | TopER without Ricci, with HKS    | 89.63±3.65  | 81.58±3.54  | 92.08±4.23  | 75.20±3.59  | 92.75±2.47   |
>
> **Time Results**
>
> | Method                                          | BZR    | COX2   | MUTAG  | PROTEINS | REDDIT-B |
> | ----------------------------------------------- | ------ | ------ | ------ | -------- | -------- |
> | TopER HKS                                       | 2.35   | 3.53   | 0.42   | 24.44    | 949.35   |
> | Spectral Zoo                                    | 14.22  | 23.99  | 8.87   | 53.04    | 398.64   |
> | TopER degree                                    | 1.25   | 2.66   | 0.27   | 21.92    | 777.85   |
> | TopER (full original with Ricci)                | 453.46 | 579.28 | 133.95 | 3003.32  | 31868.64 |

---

### Official Review · Reviewer_xFgk · 2025-06-25

**Clarity:** 3
**Significance:** 3
**Originality:** 3
**Rating:** 5
**Confidence:** 3

**Summary:**

The paper introduces TopER, an efficient graph embedding method based on topological data analysis, which simplifies the computationally expensive Persistent Homology approach by tracking the evolution rate of graph substructures during filtration. The method produces low-dimensional, interpretable 2D embeddings (pivot and growth) and is validated on a diverse of graph classification benchmarks.

**Questions:**

- How can users systematically choose the most effective filtration function for a new dataset? Are there heuristics or automated strategies?

- Could TopER be adapted for temporal graphs, where filtration sequences evolve over time?

- How does TopER perform on noisy graphs? For example, graphs with noisy features or partly lacked edges.

**Ethical Concerns:**

["NO or VERY MINOR ethics concerns only"]

**Final Justification:**

This paper presents an efficient 2D graph visualization tool that could be valuable for interpreting complex real-world graphs. During the rebuttal phase, the authors addressed most of my concerns by highlighting the competitive performance of this 2D embedding method. The experiments on temporal graphs demonstrate its potential for extended applications, though additional tests on noisy graphs would further strengthen the work in the robustness of the method. Therefore, I recommend acceptance, while I have to say my confidence is limited due to my lack of deep expertise in this specific subfield.

I have reviewed Reviewer jFiC's comments and the authors' response. Some critiques are constructive, particularly: (1) A more thorough comparison with related works (_e.g._, DiffPool, SAGPool), and (2) A detailed breakdown of training time, possibly through an ablation study on each component's computational cost. On the other hand, I think the performance critiques may be overly stringent—given the method's lightweight nature, it is reasonable not to surpass more complex approaches on all benchmarks.

**Limitations:**

See Weakness and Questions

**Quality:**

3

**Strengths And Weaknesses:**

**Strength**

- The proposed TopER addresses the scalability limitations of PH, offering a practical solution for large-scale graph analysis.

- The 2D embeddings provide clear visualizations, aiding in cluster analysis and outlier detection.

**Weakness**

- The choice of filtration function can visibly impact performance, and optimal selection may require domain expertise.

- The linear regression model may oversimplify complex topological relationships, though the authors argue higher-order terms are negligible.

- The 2D embeddings, while interpretable, may lose nuanced structural details compared to higher-dimensional methods.

---

> ### Author Rebuttal · Authors · 2025-07-29
>
> *We thank the reviewer for their insightful feedback. Below, we respond to each point with new experiments and detailed clarifications that reinforce the robustness and novelty of our approach. We hope these updates address all remaining concerns, and we would be glad to clarify any further questions.*
>
> ---
> &nbsp;
>
> ## W1 & Q1. Choice of Filtration
> >The choice of filtration function can visibly impact performance, and optimal selection may require domain expertise.
>
> **Response:** Thank you for this valuable observation. We fully agree that the choice of filtration function can significantly impact performance and that domain knowledge can guide this selection. However, our goal is to design a method that performs well across diverse graph domains without relying on dataset-specific tuning.
>
> To this end, we demonstrate that general-purpose functions, such as degree, centrality, and closeness, already achieve strong individual performance (Table 5). We further provide a broadly applicable set of functions commonly used in persistent homology, and **introduce an automated selection procedure**. Specifically, we apply t-tests to identify statistically significant functions ($p < 0.05$), followed by Lasso regularization to eliminate redundant features and retain only the most informative ones (Section 4.1).
>
> As shown in Table 5, while the effectiveness of individual filtrations varies by dataset, TopER’s aggregated representation, guided by this selection process, consistently delivers strong results without requiring domain-specific choices. We believe this design strikes a balance between flexibility and practical usability, and we will clarify this point more explicitly in the revision.
>
> ---
> &nbsp;
>
> ## W2. Linear Fit
> > The linear regression model may oversimplify complex topological relationships, though the authors argue higher-order terms are negligible.
>
> **Response:** Thank you for this comment. We agree that higher-order relationships may exist in some graphs, but our experiments show that for the quantities TopER models, specifically, the growth of edges with respect to nodes during filtrations, linear fitting provides a strong and stable approximation. As shown in Appendix B.3 and Figure 6, the linear and quadratic fits are nearly indistinguishable, and **the average quadratic coefficients (Table 13) are negligible across datasets (Between 0.00002 and 0.01)**. This indicates that the underlying growth is inherently linear in the types of filtrations we use.
>
> Moreover, preserving this linearity helps ensure interpretability, robustness, and computational simplicity. That said, TopER is flexible: if future applications require fitting more complex, non-monotonic quantities, higher-order terms can be introduced without altering the framework. We will clarify this flexibility and empirical rationale in the revision.
>
> ---
> &nbsp;
>
> ## W3. 2D Embeddings losing info
> >The 2D embeddings, while interpretable, may lose nuanced structural details compared to higher-dimensional methods.
>
> **Response:** Thank you for raising this point. While we agree that a 2D embedding may not capture all the nuanced structural details that high-dimensional methods can, TopER is designed to address a different and equally important need: producing **interpretable, low-dimensional representations** that enable **graph-level visualization and insight.**
>
> There is a significant gap in graph machine learning when it comes to effective graph-level visualization tools. TopER directly addresses this by providing compact, meaningful 2D embeddings that reveal structural patterns, class clusters, and outliers across datasets. Despite its simplicity, TopER also achieves **competitive classification performance**, demonstrating that interpretability does not have to come at the cost of utility.
>
> In the revision, we will more explicitly highlight TopER’s value as both a visualization tool and a lightweight embedding method that complements more expressive models.
>
> ---
> &nbsp;
>
> ## Q2. Temporal Graphs?
> >Could TopER be adapted for temporal graphs, where filtration sequences evolve over time?
>
> Thank you for this insightful suggestion. TopER naturally extends to temporal graphs by treating each filtration step as a 2D embedding: as the graph evolves, **these embeddings trace a trajectory in the plane that captures temporal changes**. Building on this idea, we have developed a temporal TopER variant (*TopERTemporal*) for discrete‐time graph property prediction.
>
> Our preliminary experiments on eight benchmark datasets (including blockchain and social networks) show that TopERTemporal significantly outperforms both static and temporal baselines:
>
> | Method      | Average ROC-AUC |
> |-------------|------------------|
> | GIN         | 0.4754           |
> | TDA-GIN     | 0.5314           |
> | EvolveGCN   | 0.7772           |
> | GRUGCN      | 0.7826           |
> | HTGN        | 0.7658           |
> | GraphPulse  | 0.8575           |
> | **TopERTemporal**     | **0.9416**           |
>
> We are preparing a separate submission detailing this extension. In the revised conclusion of the current manuscript, we will include a brief discussion of these results and their implications. This work underlines TopER’s versatility and its potential for a wide range of temporal‐graph learning tasks.
>
> ---
>
> &nbsp;
>
> ### Q3. Noisy Graphs?
> > How does TopER perform on noisy graphs? For example, graphs with noisy features or partly lacked edges.
>
> We thank the reviewer for raising this important question. TopER is fundamentally grounded in topological descriptors, and topology is well-known to be robust against small perturbations, such as the addition or deletion of a few edges or nodes. Intuitively, this is because the shape of the filtration sequence (i.e., how subgraphs evolve across thresholds) tends to remain stable under minor changes.
>
> Under an edge insertion as perturbation noise scenario, the change in regression parameters is bounded (change in slope for an added edge is $$|b' - b| \le \frac{|x_j - \bar{x}|(1 - \frac{1}{n})}{\sum_{i=1}^n (x_i - \bar{x})^2}$$ and inversely proportional to the variance of node counts across filtration steps, which is typically nontrivial in practice. Therefore, TopER is provably stable under small structural perturbations, and we already support this claim in the paper via Theorem 3.4 and Corollary 3.5, which bound changes in TopER vectors in terms of the Wasserstein distance between filtrations or the $L^1$ norm of filtration functions.

---

> > ### Comment · Reviewer_xFgk · 2025-08-04
> >
> > Thank you for your careful response—it resolved my concerns. This method seems valuable for interpretable graph data visualization and demonstrates strong performance across various benchmarks. I’ll maintain my 5 rating, though with slightly lower confidence due to my limited familiarity with this subfield.

---

> > > ### Author Response · Authors · 2025-08-04
> > >
> > > Thank you very much for your comments. If you have further questions, we'd be more than happy to clarify them.
> > >
> > > Best wishes,
> > >
> > > Authors

---

### Official Review · Reviewer_jFiC · 2025-07-03

**Clarity:** 2
**Significance:** 2
**Originality:** 4
**Rating:** 4
**Confidence:** 4

**Summary:**

This paper proposed TopER, a topology-aware graph representation learning framework, that overcomes the efficiency problem of persistence homology by modeling the evolution of topological filtration, rather than actual persistence features. TopER generates low-dimensional, interpretable embeddings for whole graph by fitting a linear model on evolution of the graph. The method is evaluated on graph classification, clustering, and visualization tasks across several datasets, and is shown to be competitive, or close to state-of-the-art baselines.

**Questions:**

- How does the popularity filtration differ from PageRank? Could you include a discussion?
- Are the report runtimes in lines 244-253 end-to-end, including precomputing filtration functions, or just the embedding and classification? How do these runtime compares to GPU-accelerated GNNs and pooling methods?
- Does TopER scale better than Spectral Zoo?
- In practice, does TopER's interpretability justify the performance loss in some datasets? What are the idea use-cases for TopER?

**Ethical Concerns:**

["NO or VERY MINOR ethics concerns only"]

**Final Justification:**

I found the core algorithm in this paper to be original and well-motivated from the beginning. However, I had concerns about its performance on some benchmarks and the high runtime, particularly due to the use of Ollivier-Ricci curvature. After an extended back-and-forth during the discussion phase, the authors introduced a variant of their method using the heat kernel signature (HKS) as a lighter filtration function. This update led to meaningful runtime reductions while preserving or even improving performance on several datasets. I’m now satisfied with the method's practicality and robustness, and have updated my score accordingly.

**Limitations:**

The authors have included limitations of their work.

**Quality:**

1

**Strengths And Weaknesses:**

### Strengths
The paper is well written and easy to follow. The paper identifies and addresses the lack of interpretability in high-dimensional data and graph embeddings, as well as the shortcomings of existing topological methods. Experiments are thorough and the paper provides ablation studies to showcase its performance. The 2D embeddings computed by TopER offer interpretability, allowing for exploratory data analysis, which is rarely a focus in other graph representation learning papers.

### Weaknesses
- From lines 27-31, the paper criticizes computational cost of pooling node embeddings in graphs and motivated direct graph embeddings, but does not mention DiffPool [1], SAGPool [2], and related recent pooling-based methods that address this problem. These works should at least be acknowledged as prior work in producing graph-level representations.
- The phrase "reducing computational complexity" is over-used throughout the paper even though the reported runtimes seem too high for a relatively small graph (e.g. almost 1.5 days for Reddit-5k). I doubt on the claimed efficiency of TopER, especially when compared to graph neural network and related methods that can be GPU accelerated.
- While DiffPool is included in the baselines, the results are not reported in Table 3. Furthermore, the authors need to compare more recent pooling methods like EdgePool [2], SAGPool.
- On several benchmarks, TopER underperforms, but the paper does not adequately discuss why or what settings it excels in. The average deviation metric (“Avg$\downarrow$” in Table 3) may obscure poor performance on some datasets and is somewhat misleading.
- On lines 368–369, it is claimed that TopER homogenizes graph representations in a way that existing models cannot. However, models like GPSE [3] and GFSE [4] are designed to do exactly this by being pre-trained on a large dataset from different sources, this claim might be an overstatement.

[1] Ying, Zhitao, et al. "Hierarchical graph representation learning with differentiable pooling." Advances in neural information processing systems 31 (2018).

[2] Lee, Junhyun, Inyeop Lee, and Jaewoo Kang. "Self-attention graph pooling." International conference on machine learning. pmlr, 2019.

[3] Cantürk, Semih, et al. "Graph positional and structural encoder." arXiv preprint arXiv:2307.07107 (2023).

[4] Chen, Jialin, et al. "Towards A Universal Graph Structural Encoder." arXiv preprint arXiv:2504.10917 (2025).

---

> ### Author Rebuttal · Authors · 2025-07-29
>
> *We thank the reviewer for their insightful feedback. Below, we address each concern with new experiments and detailed clarifications that underscore our approach’s robustness and originality. We hope these revisions resolve any remaining questions and respectfully request your reevaluation in light of the additional evidence.*
>
> ---
> &nbsp;
>
> ## W1&W3. Comparison with Pooling Methods
> >...but does not mention DiffPool [1], SAGPool [2], and related recent pooling-based methods that address this problem
>
> **Response:** Thank you for this helpful suggestion. We have compared TopER with the mentioned methods. As shown in the table below, our model consistently outperforms or matches the performance of recent pooling-based methods such as DiffPool and SAGPool across a range of benchmarks. These results further support the effectiveness of our direct-embedding approach for graph-level representation.
>
> We agree that DiffPool [1], SAGPool [2], and related methods should be acknowledged. In the revision, we will add a discussion of these approaches in the Related Work section and include their results in the main comparison table.
>
> | Method     | BZR   | COX2  | MUTAG | PROTEINS | IMDB-B | IMDB-M | REDDIT-B |
> |------------|-------|-------|--------|----------|--------|--------|-----------|
> | Top-K      | 79.40 | 80.30 | 67.61  | 69.60    | 73.17  | 48.80  | 79.40     |
> | MinCutPool | 82.64 | 80.07 | 79.17  | 76.52    | 70.77  | 49.00  | 87.20     |
> | DiffPool   | 83.93 | 79.66 | 79.22  | 73.63    | 68.60  | 45.70  | 79.00     |
> | SAGPool    | 82.95 | 79.45 | 76.78  | 71.86    | **74.87**  | 49.33  | 84.70     |
> | **TopER**   | **90.13** | **82.01** | **90.99**  | **74.58**    | 73.20  | **50.00**  | **92.70**     |
>
>
> ---
>
> &nbsp;
>
> ## W2. Computational Time
> >The phrase "reducing computational complexity" is over-used throughout the paper even though the reported runtimes seem too high..
>
> **Response:** We thank the reviewer for highlighting this important concern regarding computational time. We agree that the phrase *reducing computational complexity* should be used more precisely. While TopER itself is computationally lightweight, with linear complexity in the number of filtration steps and graph size ($O(n(|V| + |E|))$), the primary runtime bottleneck arises from the computation of certain filtration functions, most notably the Ollivier-Ricci curvature, which is known to be computationally demanding and is external to our core method.
>
> To clarify this point, we reran our pipeline on the Reddit-5K and MolHIV datasets, excluding Ollivier-Ricci curvature. This reduced the total runtime **from 1.5 days to 6 hours and 20 minutes** on Reddit-5K, and **from 16.5 hours to 3 hours and 1 minute** on MolHIV (40K graphs), using the same hardware specified in the paper.
>
> Moreover, as shown in Figure 5 of the Appendix, TopER remains highly scalable: when using simple degree-based filtrations, it computes embeddings for a *100K-node graph in under 50 seconds*. In our revision, we will clarify this distinction throughout the paper, emphasizing that TopER’s embedding pipeline is efficient, while the choice of filtration function can significantly impact preprocessing time.
>
>
> ---
>
> &nbsp;
>
> ## W4. Classification Performance
>
> > On several benchmarks, TopER underperforms, but the paper does not adequately discuss why or what settings it excels in...
>
> **Response:** Thank you for the insightful comment. While TopER may not lead on every benchmark, it achieves the best performance across diverse graph domains despite using very few parameters. Our main goal is not maximizing classification accuracy, **but producing low-dimensional, interpretable embeddings** that are useful for **visualization** and can serve as **effective features for downstream GNNs**.
>
> In general, TopER shows strong performance on datasets where structural graph features (e.g., degree, closeness, Ricci curvature) are informative and discriminative across classes. This includes molecular and social datasets where meaningful topological signals can be captured through filtration. On datasets with very subtle class differences or noisy graph structures (e.g., IMDB-M), where node attributes or learned representations may be more advantageous, TopER’s performance is understandably more limited. This is the main analysis on TopER performance, and we will include it in our discussion. Thank you for your suggestion.
>
> Regarding the average deviation metric ("Avg" in Table 3), we agree that it may mask underperformance on individual datasets. However, we include it to provide a compact, high-level comparison across many baselines and tasks. To avoid confusion, we will revise the manuscript to clarify this intent. Also, we will highlight cases where TopER does not perform best, and discuss dataset-specific characteristics (e.g., size, attribute availability) that may impact performance.
>
>
> ---
>
> &nbsp;
>
> ## W5. Missing Comparisons
>
> > ... However, models like GPSE [3] and GFSE [4] are designed to do exactly this...
>
> **Response:** We appreciate the reviewer’s attention to recent developments and thank them for pointing out GPSE [3] and GFSE [4]. We agree that both works contribute toward universal graph representations by leveraging large-scale pretraining. However, TopER pursues a distinct design objective: our goal is not universal pretraining but the construction of low-dimensional, interpretable, and visualizable embeddings, directly grounded in topological intuition.
>
> While models like GPSE and GFSE focus on transferability and scalability via large encoders and massive training corpora, TopER emphasizes simplicity and transparency. For example, TopER produces two-dimensional embeddings for each graph, which makes it uniquely suited for visual comparison and anomaly detection without relying on dimensionality reduction.
>
> On graph visualization specifically, we would like to clarify that [3] (GPSE) visualizes graphs only after applying PCA to high-dimensional representations. This differs fundamentally from TopER’s direct 2D outputs, where the axes themselves are meaningful and interpretable (pivot and growth). As for [4] (GFSE), it was released on arXiv on April 15, during the NeurIPS submission period, and does not present any graph visualizations. Furthermore, we note that [4] is a very interesting work, but it has not yet been finalized yet, having different versions for NeurIPS and ICLR. We will be citing this work in our related articles.
>
> To address the reviewer’s concern, we will revise the manuscript to soften the claim on lines 368–369. Rather than stating that existing models cannot homogenize representations, we will clarify that existing models do not provide low-dimensional, interpretable embeddings suitable for direct graph-level visualization across datasets, which remains a unique strength of TopER.
>
> Finally, while the GFSE code is not publicly available, we were able to run comparisons with GPSE. Across standard benchmarks, TopER consistently outperforms GPSE. We will include these results in the main comparison table in our revision.
>
>
> |           | BZR              | COX2             | MUTAG            | PROTEINS         | IMDB‑B           | IMDB‑M           |
> | --------- | ---------------- | ---------------- | ---------------- | ---------------- | ---------------- | ---------------- |
> | **GPSE**  | 80.49 ± 4.18     | 78.37 ± 2.62     | 87.19 ± 8.66     | 72.15 ± 3.66     | 69.30 ± 3.61     | 47.40 ± 5.40     |
> | **TopER** | **90.13 ± 4.14** | **82.01 ± 4.59** | **90.99 ± 6.64** | **74.58 ± 3.92** | **73.20 ± 3.43** | **50.00 ± 4.02** |

---

> ### Comment · Reviewer_jFiC · 2025-08-04
>
> Thanks for the response and clarification. While the added comparisons and clarifications are helpful, some concerns remain:
>
> - The explanation for TopER's underperformance on certain datasets still feels surface-level. Simply attributing poor results to “noisy graph structures” or “subtle class differences” is vague. I would encourage a more systematic analysis—e.g., do performance trends correlate with graph size, density, or attribute sparsity?
>
> - On the claim about homogenizing graph representations, I find the distinction between TopER and GPSE/GFSE somewhat overstated. Even if GPSE uses PCA for visualization and TopER directly outputs 2D embeddings, both aim to enable meaningful cross-graph comparisons. The paper should reframe this contrast more carefully rather than implying that other methods cannot achieve this.
>
> I noticed that several of my original questions remain unanswered, particularly regarding the distinction between popularity filtration and PageRank and scalability relative to Spectral Zoo.
>
> Lastly, what is the impact on performance after removing Ollivier-Ricci curvature for speeding up the model?
>
> While the authors have addressed some concerns, several key questions remain unanswered, and some claims still feel overstated. I appreciate the additions, especially the new comparisons, but a clearer discussion of TopER’s limitations, practical use-cases, and performance trade-offs is still needed to fully assess the contribution.

---

> > ### Comment · Area_Chair_LsVo · 2025-08-06
> >
> > Thank you for contributing to the discussion. Could you please review the additional responses provided by the authors and share your thoughts or engage with them further?

---

> ### Author Response · Authors · 2025-08-04
> **Answers to Questions**
>
> Dear Reviewer jFic,
>
> Thank you for your comments. We just realized that our earlier reply addressed only the listed weaknesses, and we unintentionally omitted responses to the additional questions. We are including those responses below and will provide further clarifications for any new questions shortly.
>
> ---
>
> ## Q1. Popularity vs. PageRank
> >How does the popularity filtration differ from PageRank? Could you include a discussion?
>
> **Response:** Thank you for the question. Given our definition of the popularity function, the distinction from PageRank is important to clarify, and we will add a concise comparison in the revision.
>
> Our popularity filtration is a local, two-hop refinement of degree: for each node vv,
> $\mathcal{P}(v)=\deg(v)+\frac{1}{|\mathcal{N}(v)|}\sum_{u\in\mathcal{N}(v)}\deg(u),$
> which augments a node’s own connectivity with the average degree of its immediate neighbors. The intent is to emphasize influence through association with well-connected neighbors; it remains a purely local statistic (involving the 1- and 2-neighborhood) and yields a filtration sequence that reflects how this localized “popularity” evolves.
>
> PageRank, in contrast, is a global diffusion-based centrality measure. It assigns importance via the stationary distribution of a random walk, recursively propagating influence across paths of arbitrary length. Thus, PageRank captures long-range, global structure, whereas popularity captures immediate neighborhood quality without iterative propagation or normalization over the whole graph.
>
> These two notions are complementary: popularity is inexpensive, interpretable, and focused on local neighborhood strength (like upgraded degree function), while PageRank encodes global reachability and recursive authority. In the revised paper we will include this comparison, and, if helpful, can also add an ablation or empirical evaluation where PageRank is used as an additional filtration function to contrast or combine it with popularity.
>
> ---
>
> ## Q2. Reported runtimes
> >Are the report runtimes in lines 244-253 end-to-end, including precomputing filtration functions, or just the embedding and classification? How do these runtime compares to GPU-accelerated GNNs and pooling methods?
>
> **Response:** Thank you for raising this timing concern. The reported runtimes (lines 244–253) are end-to-end: precomputing filtration functions (including the expensive Ollivier-Ricci curvature), producing TopER embeddings, and training the lightweight MLP. TopER’s core embedding is efficient, linear in graph size and filtration steps $O(n(|V|+|E|))$, and highly scalable: with simple degree filtrations it processes a 100K-node graph in under 50 seconds (Fig. 5). The main bottleneck is the *optional Ollivier-Ricci computation*. Removing it reduces Reddit-5K time **from 1.5 days to 6h20m** and MolHIV (40K graphs) **from 16.5h to 3h1m** on the same CPU hardware. Also, the classification itself is cheap (under 5 minutes on Reddit-5K).
>
> While not directly comparable due to CPU vs GPU, common GPU-trained GNNs and pooling methods run in similar orders of magnitude, suggesting the full pipeline is reasonable, especially when costly filtrations are omitted or approximated. We will clarify the distinction between the lightweight TopER embedding and the optional expensive filtration choices in the revision and include this discussion explicitly.
>
> ---
>
> ## Q3. Scalability of Toper vs. Spectral Zoo
> >Does TopER scale better than Spectral Zoo?
>
> Yes. Spectral Zoo builds rich global descriptors using spectral computations such as eigendecompositions and heat kernel signatures, which involve costly linear algebra. Even approximate methods typically incur superlinear cost in the number of nodes, making them relatively heavy on large graphs.
>
> TopER, by contrast, uses local or near-local filtration statistics and scales linearly with graph size and edges, $O(n(|V| + |E|))$. With simple filtrations (e.g., degree), embedding a 100K-node graph takes under a minute on CPU (Fig. 5). The only expensive steps, like Ollivier-Ricci, are optional; the core pipeline is lightweight and efficient. Classification is also fast.
>
> While the two approaches are complementary, TopER offers a clear scalability advantage, particularly for large or resource-constrained settings. We will add a brief comparison to Spectral Zoo in the revision.

---

> ### Author Response · Authors · 2025-08-04
> **Questions ctd.**
>
> ## Q4. TopER's interpretability, and practical use
>
> > In practice, does TopER's interpretability justify the performance loss in some datasets? What are the idea use-cases for TopER?
>
> **Response:** Yes. In many practical settings TopER’s interpretability outweighs modest performance gaps, and sometimes it even matches or beats opaque baselines. The sample figures in the paper illustrate this: the pivot/growth coordinates give low-dimensional, deterministic descriptors that cleanly separate classes, expose outliers, and support visualization of clusters. That transparency lets users understand why a graph is classified a certain way (e.g., abnormal edge growth vs node expansion with respect to the used filtration function) and compare structures both within and across datasets in a topology-inspired way.
> Ideal use cases for TopER include:
>
> * Fast, scalable prefiltering or clustering of graphs before applying heavier models
> * Explainable decision support in scientific domains (biology, chemistry, social networks) where structural drivers matter
> * Domain transfer and cold start since no training is required
> * Anomaly or change detection via shifts in filtration evolution over time
> * Resource-constrained or large-scale scenarios where spectral or deep GNN methods are too expensive
> * Hybrid systems, where TopER features augment learned models and serve as interpretable anchors
>
> When peak accuracy on a specific benchmark is the only goal, more expressive learned or spectral methods may outperform TopER; in those cases, TopER remains valuable as a **diagnostic, initialization, or comparative analysis tool**. We will add a concise paragraph in the revision summarizing these trade-offs and point to the provided figures as concrete evidence of its interpretability benefits.
>
> ---
>
> Thank you again for your feedback and time. We will respond to your additional questions shortly.

---

> ### Author Response · Authors · 2025-08-04
> **Responses to Follow-up Questions**
>
> Dear Reviewer jFiC,
>
> We thank you for your thoughtful follow-up and the opportunity to further clarify our work. Below, we address the remaining concerns point by point.
>
> ---
>
> ## FQ1. On the explanation of underperformance on certain datasets
> > The explanation for TopER's underperformance on certain datasets still feels surface-level
>
> Thank you for the thoughtful critique and for pushing us beyond high-level explanations. While TopER does not beat every baseline on every dataset, its average deviation from the best performance is the lowest (Table 3), highlighting robust and consistent performance across diverse domains, from molecular graphs to social networks. To make the underperformance cases more precise, in the revision, we will include a systematic analysis. Preliminary observations show that TopER performs best when structural signals are clean and informative (e.g., REDDIT-B and MUTAG, where degree and Ricci-based patterns separate classes well) and degrades on graphs like IMDB-M that lack node attributes and exhibit noisy or overlapping class structure. Performance also tends to correlate positively with sparsity and clear filtration-growth dynamics. We will include a detailed analysis correlating normalized accuracy (relative to SOTA) with graph size, average degree, degree heterogeneity (variance/skew), and average clustering coefficient to make these trends explicit. Thank you again for encouraging a more systematic investigation.
>
> ---
>
> ## FQ2.  Homogenization of graph representations and comparison with GPSE.
> > On the claim about homogenizing graph representations, I find the distinction between TopER and GPSE/GFSE somewhat overstated.
>
> We agree that GPSE and GFSE contribute toward homogenizing graph representations; they are also recent and reflect the growing interest among researchers towards homogenizing graph datasets. TopER has similar goals. Our intended distinction was not to claim that existing models cannot compare graphs across datasets, but rather that they do not produce low-dimensional embeddings where *each axis is interpretable*. Unlike these methods, TopER’s two coordinates (pivot and growth) are derived from an explicit, meaningful filtration process. We studied this interpretability in Appendix B4 and Appendix Figure 7 shows the example interpretations that we can make from the TopER values: (_Positive pivot graphs display a cluster of vertices that are closely interconnected and appear early in the filtration process. On the other hand, negative pivot graphs feature loosely connected nodes where the edges enter the filtration at a later stage. Graphs with zero pivot are usually quasi-complete graphs._)
>
> We will revise lines 368–369 to soften the wording and make the distinction clearer. The new phrasing will highlight that, to our knowledge, TopER is unique in directly producing interpretable 2D embeddings for cross-dataset visualization without relying on learned high-dimensional encodings or opaque projections. We will add a brief discussion of these models, explicitly noting the key differences. Thank you again for bringing these important works to our attention.
>
> ---
>
> ## FQ3. On popularity filtration vs. PageRank
>
> Please see our responses to Q1 above.
>
> ---
>
> ## FQ4. On the scalability of TopER vs. Spectral Zoo scalability
>
> In addition to our original response (Q3 above), we acknowledge the oversight in not providing a direct runtime comparison; we had computed these values, which we report below, in the preliminary phases of the project. Spectral Zoo computes high-order spectral moments and requires eigendecomposition, which scales cubically with graph size. In contrast, TopER (with degree-based filtrations) has linear complexity in graph size and number of thresholds.
>
> Time cost in seconds. For our method, TopER, with degree function:
>
> |         	| BZR   | COX2  | MUTAG | PROTEINS | IMDB-B | IMDB-M | REDDIT-B | REDDIT-5K |
> |-------------|-------|-------|-------|----------|--------|--------|----------|-----------|
> | Spectral Zoo| 14.22 | 23.99 | 8.87  | 53.04	| 45.52  | 68.73  | 398.64   | 2372.4	|
> | TopER      	| 1.25  | 2.66  | 0.27  | 21.92	| 21.65  | 25.79  | 777.85  | 2646.33  |
>
> ---
>
> ## FQ5. On performance without Ollivier-Ricci curvature
>
> Thanks again for this question. We are currently running the experiments without O.Ricci, and will post the results soon.

---

> ### Author Response · Authors · 2025-08-04
> **TopER without O.Ricci Table**
>
> ## FQ5. On performance without Ollivier-Ricci curvature
>
> Below, we give the TopER results without O. Ricci. As the table shows, removing O.Ricci features leads to a ~3–4 point drop on molecular datasets (e.g., BZR, COX2, MUTAG), but has minimal effect (<1 point) on larger social benchmarks like PROTEINS, REDDIT, and IMDB. This suggests Ricci features are most helpful when local structure is key, while TopER’s other descriptors suffice for large, sparse graphs. Since Ricci is the main runtime cost, it can be omitted for faster inference with limited accuracy loss.
>
>
> | Model  |    BZR       |    COX2       |   MUTAG       |  PROTEINS     |  IMDB-B       |  IMDB-M       |  REDDIT-B     | REDDIT-5K     |
> |--------|--------------|---------------|---------------|---------------|---------------|---------------|---------------|---------------|
> | TopER  | 90.13±4.14| 82.01±4.59   | 90.99±6.64    | 74.58±3.92    | 73.20±3.43    | 50.00±4.02    | 92.70±2.38| 56.51±2.22    |
> | TopER no O.Ricci  | 87.00±4.30| 77.96±8.38   | 87.78±7.84   | 74.04±3.86    | 73.50±3.53    | 50.00±5.44    | 91.90±2.63| 56.37±1.89    |
>
> ---
> Thanks again for your valuable feedback. We'd be happy to clarify if you have further questions.

---

> > ### Comment · Reviewer_jFiC · 2025-08-07
> >
> > Thank you for the detailed responses and additional experiments. While I appreciate the authors' effort in addressing each point, I remain unconvinced by the claims around efficiency and practical utility.
> >
> > Despite emphasizing TopER’s lightweight nature, the method’s runtime remains high, especially when using Ollivier-Ricci curvature. Omitting it does reduce computational cost significantly, but this also leads to notable drops in performance. For example, on BZR and MUTAG, TopER no longer ranks first or even second. This undermines the argument that the method is both efficient and competitive.
> >
> > I also remain skeptical of the idea that underperforming models are suitable for visualization simply because they produce interpretable embeddings. Strong performance is still important for visualizations to be meaningful and trustworthy. Moreover, while TopER emphasizes axis interpretability, similar post-hoc analyses could potentially be applied to GPSE or related models to interpret dimensions in their embedding spaces.
> >
> > Overall, the method introduces an interesting direction, but I feel the claims around efficiency and robustness across benchmarks are still somewhat overstated.

---

> ### Author Response · Authors · 2025-08-07
>
> Dear Reviewer jFiC,
>
> We value your comments, especially as you rated our work *4: Excellent for originality*. We would like to convince you of the strength of our results as well.
>
> When we began our experiments, we assumed O.Ricci would be asked for (being widely used filtration function in PH) and designed our model accordingly. However, the selection of filtration functions are flexible, TopER can operate with cheaper functions. For example, upon your review, we implemented the old and fast *heat kernel signature* (HKS) as a filtration function [1], and this new variant without Ricci even yielded better results than the original TopER article version on three datasets while significantly reducing time costs. We offer this as evidence that TopER is efficient and scalable.
>
> Furthermore, we are reporting these results just hours after your comment, whereas our competitors would typically require several days to adjust their pipelines and report results.
>
> As a final note, we found GPSE to be a very interesting method, and we will cite it. However, the modifications you mention to bring it into TopER comparisons have not been reported anywhere and would likely require new research articles to define and evaluate properly. Meanwhile, we hope our findings help demonstrate that TopER is both practical and adaptable, even without complex filtration steps. Please let us know if you have further questions or comments.
>
> **Performance Results**
>
> | Model                            | BZR         | COX2        | MUTAG       | PROTEINS    | REDDIT-B     |
> |----------------------------------|-------------|-------------|-------------|-------------|--------------|
> | SOTA                             | 89.00±5.00  | 85.53±1.60  | 92.63±2.58  | 77.30±0.89  | 91.03±0.22   |
> | TopER with Ricci                 | 90.13±4.14  | 82.01±4.59  | 90.99±6.64  | 74.58±3.92  | 92.70±2.38   |
> | TopER without Ricci, with HKS    | 89.63±3.65  | 81.58±3.54  | 92.08±4.23  | 75.20±3.59  | 92.75±2.47   |
>
> **Time Results**
>
> | Method                                         | BZR    | COX2   | MUTAG  | PROTEINS | IMDB-B | IMDB-M | REDDIT-B  | REDDIT-5K |
> |-----------------------------------------------|--------|--------|--------|----------|--------|--------|-----------|------------|
> | TopER HKS                                     | 2.35   | 3.53   | 0.42   | 24.44    | —      | —      | 949.35    | —          |
> | Spectral Zoo                                  | 14.22  | 23.99  | 8.87   | 53.04    | 45.52  | 68.73  | 398.64    | 2372.4     |
> | TopER only degree                                  | 1.25   | 2.66   | 0.27   | 21.92    | 21.65  | 25.79  | 777.85    | 2646.33    |
> | TopER (full original with Ricci)              | 453.46 | 579.28 | 133.95 | 3003.32  | —      | —      | 31868.64  | —          |
>
> **Missing results will be added if the deadline allows.
>
> [1] Sun, J., Ovsjanikov, M. and Guibas, L., 2009, July. *A concise and provably informative multi‐scale signature based on heat diffusion*. In Computer graphics forum.

---

> ### Author Response · Authors · 2025-08-08
> **Complete tables**
>
> Dear Reviewer jFiC,
>
> Here are the complete tables from our previous message, with the best TopER performance in bold. We observe that replacing O. Ricci with HKS significantly reduces computation time while maintaining or even improving performance. Please let us know if you have any further questions or feedback.
>
> **Accuracy results**
> | Model                         |            BZR |           COX2 |          MUTAG |       PROTEINS |         IMDB-B |         IMDB-M |       REDDIT-B |      REDDIT-5K |
> | ----------------------------- | -------------: | -------------: | -------------: | -------------: | -------------: | -------------: | -------------: | -------------: |
> | SOTA                          |     89.00±5.00 |     85.53±1.60 |     92.63±2.58 |     77.30±0.89 |     75.08±0.31 |     52.81±0.31 |     91.03±0.22 |     56.75±0.18 |
> | TopER without O.Ricci, with HKS |     89.63±3.65 | 81.58±3.54 | **92.08±4.23** | **75.20±3.59** | **75.00±3.49** | **50.67±5.58** | **92.75±2.47** | **57.33±2.02** |
> | TopER (original with O.Ricci )                       | **90.13±4.14** |     **82.01±4.59** |     90.99±6.64 |     74.58±3.92 |     73.20±3.43 |     50.00±4.02 |     92.70±2.38 |     56.51±2.22 |
>
> ---
>
> **Times** in seconds
>
> Method                                         | BZR    | COX2   | MUTAG  | PROTEINS | IMDB-B | IMDB-M | REDDIT-B  | REDDIT-5K |
> |-----------------------------------------------|--------|--------|--------|----------|--------|--------|-----------|------------|
> | Spectral Zoo                                  | 14.22  | 23.99  | 8.87   | 53.04    | 45.52  | 68.73  | 398.64    | 2372.4     |
> | TopER without O.Ricci, with HKS                                 | 2.35   | 3.53   | 0.42   | 24.44    |  24.1      |  29.55      | 949.35    |  5446.29         |
> | TopER (original with O.Ricci)              | 453.46 | 579.28 | 133.95 | 3003.32  | 1531.69      | 3388.35      | 31868.64  |  124565.42          |

---

> ### Comment · Reviewer_jFiC · 2025-08-08
>
> Thank you for the additional results. The inclusion of HKS as a more efficient filtration function helps address my concerns about runtime and expressiveness, without changing the original algorithm, which I really liked. Based on these updates, I am updating my score. Please ensure that the final version includes the revised performance discussion, comparisons, and runtime breakdowns. Good luck!

---

> ### Author Response · Authors · 2025-08-08
> **Thank you**
>
> Dear Reviewer jFiC,
>
> Thank you very much for your follow-up and for updating your score. We’re glad the HKS results addressed your concerns and will ensure the final version includes the revised discussion, comparisons, and runtime breakdowns.
>
> Best regards,
>
> Authors

---

### Author Response · Authors · 2025-08-03
**Rebuttal follow-up**

Dear Reviewers,

Thank you again for your insightful reviews and for engaging with our submission. We wanted to kindly follow up, as there are only a few days left in the author-reviewer discussion period. If you have any additional questions or concerns after reading our rebuttal, we would be happy to clarify or provide further information.

We truly appreciate your time and consideration.

Best regards,

The Authors

---

### Author Response · Authors · 2025-08-06
**Rebuttal follow-up**

Dear Reviewers,

Thank you again for your insightful reviews and for engaging with our submission. We wanted to kindly follow up, as there are only a few days left in the author-reviewer discussion period. If you have any additional questions or concerns after reading our rebuttal, we would be happy to clarify or provide further information. We truly appreciate your time and consideration.

Best regards,

The Authors

---

### Author Response · Authors · 2025-08-08

Dear Reviewers 2YLF and jFiC,

As today is the final day of the author–reviewer discussion period, we wanted to kindly follow up on our earlier clarifications. Please let us know if they address your concerns or if there is anything further we can provide before the discussion closes. If the clarifications resolve your concerns, we would be grateful if you could consider revising your score accordingly.

Thank you very much for your valuable time and feedback.

Authors

---

### Note · Authors · 2025-08-11

Dear Reviewers, ACs, SACs, and PCs,

First, we would like to thank you for reviewing our paper and for overseeing the process. Your constructive feedback has been invaluable in strengthening both the technical content and presentation of our work.

We are grateful for the positive feedback:

* **Clear motivation**:  Addresses the lack of interpretable, low-dimensional graph embeddings, an important and underexplored challenge in graph representation learning. (jFiC, xFgk, tsuE)

* **Novel and flexible method**:  Introduces the Topological Evolution Rate framework for directly producing interpretable 2D embeddings, adaptable in filtration choice and scalable to large graphs. (jFiC, tsuE)

* **Practical utility**:  Enables intuitive visualization, anomaly detection, and clustering, complementing high-capacity models. (xFgk, tsuE)

* **Comprehensive evaluation**: Benchmarked across molecular, biological, and social datasets, with ablations, runtime studies, and comparisons to recent pooling and structural encoder methods. (jFiC, xFgk)

* **Clarity and organization**: Well-structured manuscript with detailed appendices. (xFgk, tsuE)

During rebuttal and discussion, we:

* Replaced Ollivier–Ricci curvature with Heat Kernel Signature at jFiC’s request, reducing runtimes by up to two orders of magnitude while maintaining or improving accuracy.

* Expanded comparisons to DiffPool, SAGPool, GPSE, and Spectral Zoo, clarifying accuracy–efficiency trade-offs.

* Added deeper performance analysis, correlating TopER’s strengths and weaknesses with graph structural properties.

* Clarified differences between popularity filtration and PageRank, and between TopER and universal structural encoders.

* Committed to synchronizing theorem numbering, refining theoretical presentation, and detailing the feature-selection procedure.

All these revisions will be incorporated in the camera-ready version. We believe that TopER’s unique combination of interpretability, scalability, and competitive performance makes it a valuable contribution to graph representation learning. We hope that our clarifications, new results, and the strengths recognized by multiple reviewers will lead to a favorable decision. We want to thank you again for your valuable time and feedback.

---

### Decision · Program_Chairs · 2025-09-17

**Decision:**

Accept (poster)

**Comment:**

The paper proposes a low-dimensional, interpretable topological embedding for graphs that estimates the topological evolution rate of graph substructures as a simpler, scalable alternative to persistent homology. It supports visualization and yields competitive performance on graph clustering and classification across a variety of datasets.

The authors engaged constructively and provided meaningful rebuttal updates: they replaced Ollivier-Ricci curvature with the Heat Kernel Signature, yielding up to ~100x speedup; expanded comparisons to strong baselines to clarify accuracy vs. efficiency trade-offs; added deeper analyses linking performance to graph structure; and clarified distinctions from related approaches (such as, universal structural encoders). Including these updates into the camera-ready and polishing the theoretical section (e.g., clearly stating the assumptions) will further strengthen the paper. Overall, the combination of practical interpretability, strong aggregate accuracy (despite not always surpassing every baseline on individual datasets) and improved scalability makes this work of interest to the NeurIPS community.